# Nup98 FG domains from diverse species spontaneously phase-separate into particles with nuclear pore-like permselectivity

Hermann Broder Schmidt[†], Dirk Görlich*

Department of Cellular Logistics, Max Planck Institute for Biophysical Chemistry, Göttingen, Germany

**Abstract** Nuclear pore complexes (NPCs) conduct massive transport mediated by shuttling nuclear transport receptors (NTRs), while keeping nuclear and cytoplasmic contents separated. The NPC barrier in *Xenopus* relies primarily on the intrinsically disordered FG domain of Nup98. We now observed that Nup98 FG domains of mammals, lancelets, insects, nematodes, fungi, plants, amoebas, ciliates, and excavates spontaneously and rapidly phase-separate from dilute (submicromolar) aqueous solutions into characteristic 'FG particles'. This required neither sophisticated experimental conditions nor auxiliary eukaryotic factors. Instead, it occurred already during FG domain expression in bacteria. All Nup98 FG phases rejected inert macromolecules and yet allowed far larger NTR cargo complexes to rapidly enter. They even recapitulated the observations that large cargo-domains counteract NPC passage of NTR·cargo complexes, while cargo shielding and increased NTR·cargo surface-ratios override this inhibition. Their exquisite NPC-typical sorting selectivity and strong intrinsic assembly propensity suggest that Nup98 FG phases can form in authentic NPCs and indeed account for the permeability properties of the pore.

*For correspondence: goerlich@ mpibpc.mpg.de

Present address: [†]Department of Biochemistry, Stanford University School of Medicine, Stanford, United States

Competing interests: The authors declare that no competing interests exist.

## Introduction

Cell nuclei lack protein synthesis and therefore import required proteins from the cytoplasm. Nuclei, on the other hand, supply the cytoplasm with ribosomes, mRNAs, tRNAs, and other 'nuclear products'. Nuclear pore complexes (NPCs) conduct this nucleocytoplasmic transport (Reviewed by *Brohawn et al., 2009*; *Hetzer and Wente, 2009*; *Rothballer and Kutay, 2013*). They are embedded into the nuclear envelope (NE) and equipped with a remarkable permeability barrier that suppresses an inter-mixing of nuclear and cytoplasmic contents. This barrier is freely permeable for small molecules but becomes increasingly restrictive as the size of the diffusing species approaches a limit of ≈30 kDa in mass or ≈5 nm in diameter (*Mohr et al., 2009*). Larger objects are essentially excluded from passage but may traverse the pore when bound to an appropriate nuclear transport receptor (NTR).

NTRs are in continuous circulation between nucleus and cytoplasm (reviewed in *Görlich and Kutay, 1999*; *Weis, 2003*; *Cook et al., 2007*). They select cargo molecules on one side of the NE, traverse the NPC barrier in a facilitated manner, release cargo into the destination compartment, and return to the initial compartment for the next round of transport. There are several NTR categories. Importins and exportins use coupling to the RanGTPase system for active transport against concentration gradients. With molecular masses between 90 and 140 kDa, they are relatively large in size. NTF2, the import receptor for RanGDP (*Ribbeck et al., 1998*), is smaller (≈30 kDa for the homodimer). It is structurally unrelated to importins/exportins (*Bullock et al., 1996*) but homologous to the Mex67p/Mtr2p heterodimer, which functions in RNA export (See e.g., *Strasser et al., 2000*). Hikeshi represents yet another category. It mediates nuclear import of Hsp70 (*Kose et al., 2012*).

**eLife digest** Cells of eukaryotic species—which include plants, animals, and fungi—have a nucleus that harbours the organism's genome. Two membrane layers surround the nucleus and separate its contents from the cytoplasm, where proteins are made. This separation is essential for a correct interpretation of the genetic information. Yet, various molecules, such as proteins, need to move into or out of the nucleus for the cell to work properly. This transit has to occur without an uncontrolled mixing of the contents of the nucleus and the cytoplasm happening.

Structures called nuclear pore complexes span the double membrane and allow material to be exchanged between the nucleus and the cytoplasm. Small molecules can freely pass through these complexes, while larger molecules can only be transported when bound as "cargo" to so-called nuclear transport receptors. Nuclear pore complexes are large assemblies of proteins called nucleoporins. FG nucleoporins are special in that they contain regions with a repeating pattern of two amino acids, phenylalanine ('F') and glycine ('G'). These regions are called FG domains. They bind to nuclear transport receptors and have been suspected to form a barrier that decides which molecules may pass through the nuclear pore complex. Exactly how this control is exerted has been a matter of debate.

Versions of a particular FG nucleoporin called Nup98 are found in all branches of eukaryotic life, i.e. in animals, fungi, plants, amoebas, and even in the evolutionarily most distant protozoans. When Schmidt and Görlich dispersed small amounts of Nup98 FG domains in an aqueous solution, the domains rapidly attracted each other to form 'FG particles', regardless of which species the proteins came from. These FG particles were so dense that they repelled 'normal' macromolecules, yet they allowed nuclear transport receptors, along with their bound cargoes, to rapidly enter. Taken together, the work of Schmidt and Görlich suggests that such FG particles form the transport barrier in nuclear pore complexes.

Based on these findings, Schmidt and Görlich refine a model where the FG domains form a mesh in the nuclear pore complexes that acts like a 'smart sieve'. Smaller molecules can move through gaps in the meshwork, but larger molecules are hindered. Schmidt and Görlich suggest that nuclear transport receptors help large molecules to move through nuclear pore complexes by 'melting' the FG meshwork locally, creating a path for the molecule to move through. The reconstitution of these smart barriers in the laboratory will now allow researchers to analyse the process of receptor-mediated nuclear pore passage in unprecedented (mechanistic) detail.

The NPC barrier has a remarkably high capacity, supporting up to 1000 facilitated translocation events or a mass flow of 100 MDa per pore per second (*Ribbeck and Görlich, 2001*). Facilitated NPC passage can be completed within a few milliseconds, at least when smaller cargoes are transported (*Yang et al., 2004*; *Kubitscheck et al., 2005*; *Tu and Musser, 2011*). Facilitated translocation per se does not consume metabolic energy (*Weis et al., 1996*; *Schwoebel et al., 1998*; *Englmeier et al., 1999*; *Ribbeck et al., 1999*). Instead, it is based on a higher permeability of NPCs for NTRs as compared to inert molecules (*Ribbeck and Görlich, 2001*). NPCs become active and highly efficient cargo pumps, when cargo-release from NTRs is enforced at destination, which is typically achieved by the RanGTPase system (*Görlich et al., 2003*). The NPC barrier is, however, not just an obstacle to overcome. It is also essential for any active directional transport because it prevents a dissipation of the transport-driving RanGTP-gradient and retains already transported cargoes at destination.

Facilitated translocation is evident for objects of a wide range of sizes. NTF2 (diameter: ≈5 nm) exemplifies a small translocation species. It traverses NPCs 100 times faster than the equally sized GFP (*Ribbeck and Görlich, 2001*; *Mohr et al., 2009*). 60S ribosomal subunits (diameter: ≈25 nm) are on the other end of the allowed size spectrum. Such large cargoes require multiple NTR molecules for efficient NPC passage. At least three shuttling NTRs (Xpo1/CRM1, Xpo5, and Gle2/ Rae1) contribute to nuclear export of human 60S ribosomal subunits (*Wild et al., 2010*), while Xpo1/CRM1, the Mex67 Mtr2 dimer, Arx1, and Gle2 synergise in exporting *S. cerevisiae* 60S particles (*Ho et al., 2000*; *Bradatsch et al., 2007*; *Yao et al., 2007*; *Occhipinti et al., 2013*).

This NTR-cooperation effect is also evident for smaller cargoes. Nuclear import of an IBB-2xMBP-M3 fusion (88 kDa), for example, occurs ≈10 times faster if it is simultaneously bound by two importin

molecules rather than just a single one (*Ribbeck and Görlich, 2002*). Likewise, single molecule tracking revealed that multiple importin molecules are required for transporting a ≈500 kDa M9-β-Gal tetramer through NPCs (*Tu et al., 2013*).

FG nucleoporins (Nups) are critical for NPC function (*Strawn et al., 2004*). They comprise FG domains with often hundreds of residues and up to 50 FG (phenylalanine–glycine) dipeptide motifs. Vertebrate NPCs accommodate ≈11 different FG Nups, which together should contribute ≈13 MDa of FG domain mass and >5000 FG motifs per pore (*Ori et al., 2013*).

Nup98 (*Radu et al., 1995*; *Powers et al., 1997*) is an FG Nup of special interest for this study. In its canonical form, it is initially produced as a fusion protein with the NPC scaffold component Nup96. An auto-proteolytic domain in Nup98 subsequently cleaves the fusion (*Rosenblum and Blobel, 1999*) and thereafter anchors Nup98 to the NPC scaffold (*Griffis et al., 2003*). Mammalian Nup98 comprises an FG domain of about 500 residues, including a binding site for the mRNA export mediator Gle2/ Rae1 (GLEBS domain; *Bailer et al., 1998*; *Pritchard et al., 1999*). Nup98 has been estimated to occur in ≈48 copies per NPC (*Ori et al., 2013*). The yeast *Saccharomyces cerevisiae* possesses three Nup98 paralogs, Nup145, Nup100, and Nup116 (*Wente et al., 1992*; *Wente and Blobel, 1994*), whereby only Nup145 is produced as a fusion protein and only Nup116 contains a GLEBS domain. The FG domains of Nup100 (570 residues) and Nup116 (740 residues) are considerably longer than that of Nup145N (200 residues).

FG domains are intrinsically disordered (*Dosztanyi et al., 2005*; *Denning and Rexach, 2007*) and have therefore been appearing 'featureless' on high-resolution cryo-electron tomography averages (*Beck et al., 2004*). Nevertheless, they engage in critical interactions. First, FG motifs are binding sites for NTRs (*Iovine et al., 1995*; *Rexach and Blobel, 1995*) and this interaction is a prerequisite for any facilitated translocation (*Bayliss et al., 1999*). Second, certain FG domains also bind constituents of the NPC scaffold (*Patel et al., 2007*; *Schrader et al., 2008*; *Andersen et al., 2013*; *Xu and Powers, 2013*). 'Cohesive' interactions between FG repeats themselves represent a third category. They are multivalent and primarily driven by the hydrophobicity of the FG motifs (*Ribbeck and Görlich, 2001*; *Frey et al., 2006*; *Patel et al., 2007*; *Frey and Görlich, 2009*; *Ader et al., 2010*; *Hülsmann et al., 2012*; *Labokha et al., 2013*; *Xu and Powers, 2013*).

Several lines of evidence suggest that FG domains actually constitute the NPC permeability barrier. First, certain genetic deletions of FG domains relax the barrier of *S. cerevisiae* nuclear pores (*Patel et al., 2007*). Second, the collapse of the permeability barrier during poliovirus infection coincides with a cleavage of FG domains (*Park et al., 2008*). Third, replacing Nup98 in *Xenopus* nuclei by a variant lacking the FG domain renders NPCs non-selectively permeable (*Hülsmann et al., 2012*). Additional striking evidence came from the observation that jellified FG domains show permeability properties very similar to those of authentic NPCs (See e.g., *Frey and Görlich, 2007*; *Labokha et al., 2013*, and below).

The perhaps greatest challenge in the field is to explain why interactions between an NTR and the stationary FG domains do no delay but instead greatly accelerate NPC passage in comparison to inert (i.e., non-FG-binding) reference molecules. Several solutions to this paradox have been proposed. The 'virtual gate' model (*Rout et al., 2003*) regards FG domains as entropic brushes that repel inert material while NTRs overcome this entropic barrier by binding to them.

The selective phase model assumes that cohesive interactions between FG repeats cross-link them into a sieve-like FG hydrogel, whose mesh size sets an upper size limit for passive NPC passage (*Ribbeck and Görlich, 2001*; *Frey and Görlich, 2007*). It further assumes that repeat–repeat contacts disengage locally and transiently when a transport receptor binds the corresponding FG motifs. This should allow the receptor to 'melt' with its cargo through the gel. Several lines of evidence support this model: first, replacing the highly cohesive FG domain of *Xenopus* Nup98 by non-cohesive (or by partially cohesive) ones resulted in non-selectively permeable NPCs (*Hülsmann et al., 2012*). This held true even if the replacing non-cohesive FG domain was fully proficient in NTR-binding. Second, numerous FG domains can indeed form hydrogels that behave like the NPC's permeability barrier, i.e. they exclude inert macromolecules but allow rapid influx of NTRs and NTR·cargo complexes (*Frey and Görlich, 2007*, *2009*; *Milles and Lemke, 2011*). A systematic analysis of *Xenopus* FG domains revealed that the Nup98 FG domain, which appears to be the most critical one for NPC function, also yielded the most selective FG hydrogel (*Hülsmann et al., 2012*; *Labokha et al., 2013*).

A critical gap in the argument relates, however, to the facts that (i) gels with a true NPC-like permselectivity were obtained only if the FG domain concentration exceeded a certain 'saturation limit',

typically around 200 mg/ml (*Frey and Görlich, 2007*) and (ii) that it was so far unclear if and how the relevant FG domains could possibly become sufficiently concentrated to form highly selective gels.

We now observed that Nup98 FG domains from animals, fungi, plants, amoebas, ciliates, and excavates (chosen to represent all major eukaryotic clades) indeed phase-separate spontaneously from aqueous solutions into FG hydrogels. Phase-separation was a matter of no more than seconds and occurred at critical concentrations ranging from 20 to 700 nM or 1 to 50 µg/ml FG domains, which corresponds to 1–35 µM FG motifs. These critical concentrations are exceeded by at least 2–3 orders of magnitude when the NPC scaffold initially recruits Nup98 molecules during NPC assembly. Notably, NTRs did not prevent the phase-separation of FG domains, even when present in larger excess than NPCs could possibly accommodate. The self-assembly process yielded micrometer-sized near-spherical FG particles with a preferred protein concentration of ≈250 mg/ml. Strikingly, these particles behaved like the predicted selective phase: they excluded inert macromolecules (such as the ≈25 kDa-sized mCherry) but at the same time allowed efficient NTR entry at rates that appeared mostly limited by mass transport within the surrounding buffer. NTRs diffused through these particles with a rate constant in the order of 0.1 µm$^2$/s, i.e. fast enough to traverse a 40-nm thick NPC barrier made of the same material within ≈10 milliseconds. With remarkable clarity, the FG particles also recapitulated the effect that a larger cargo must recruit several NTR molecules in order to efficiently pass an NPC. The simplest explanation for these parallels is that the actual NPC barrier is formed by such phase-separated FG hydrogel material. Our data further suggest that a mere binding of a translocating species to FG domains is insufficient for facilitated passage. Instead, it is consistent with the notion that facilitated translocation involves an NTR-mediated melting of obstructing meshes and that this 'melting' is restricted to the immediate vicinity of the NTRs.

## Results and discussion

One possible explanation for the observed permselectivity of NPCs is given by the hypothesis that the barrier-critical FG domains assemble into a sieve-like meshwork, whose contact points can be temporarily and locally resolved by translocating NTRs. Indeed, it was previously shown that several FG domains can jellify into hydrogel meshworks with such a 'smart sieve behaviour', i.e. they reject inert macromolecules above a certain size limit but let the same cargoes pass when bound to an appropriate NTR. However, only gels with a concentration of no less than 200 mg/ml FG domain (corresponding to 3–4 mM FG domain or 100–175 mM FG motifs) showed such selectivity. Compared to the ≈200 nM Nup98 present in a (pre-assembly) metaphase *Xenopus* egg extract (*Hülsmann et al., 2012*), this would require a ≈20,000-fold increase in concentration. Hence, we wondered if and how such high local concentration could possibly be reached.

### Expected minimal FG domain concentrations in authentic NPCs

Clearly, the recruitment of FG Nups to the NPC scaffold already represents a first concentration step. A lower limit for the resulting FG concentration can thus be estimated based on three assumptions: (i) only Nup98-type FG domains participate in the barrier, (ii) these domains occur in 48 copies per NPC (*Ori et al., 2013*), and (iii) not only dwell inside the central channel (length: 80 nm; width: 40 nm) but reach 70 nm into the nuclear and cytoplasmic space. The latter value represents the farthest distance from the NPC plane at which epitopes of an FG Nup (Nup358) had previously been detected by immuno-EM (*Walther et al., 2002*). These numbers give an accessible volume of $1.5 \times 10^{-18}$ litres and concentrations of 50 µM (2.5 mg/ml) for the FG domain and 2 mM for the FG motifs. When considering non-Nup98 FG domains as well, the lower bound increases to 6 mM FG motifs.

However, this is still far lower than the above-mentioned threshold for hydrogel selectivity. An effective hydrogel-based barrier should therefore assemble within NPCs only if some mechanism increases the local FG domain concentration further by a factor of 10–100.

### FG particle assembly by phase-separation from dilute aqueous solutions

In the simplest case, local concentrations can increase because the barrier-forming FG domains have an intrinsic propensity to self-concentrate and hence to phase-separate from aqueous solutions. However, even then the question still remains if the resulting protein-rich phase has an appropriate FG repeat concentration and structure to function as a barrier with NPC-like permselectivity. Given that these issues are central for our understanding of NPC function, we decided to address them systematically.

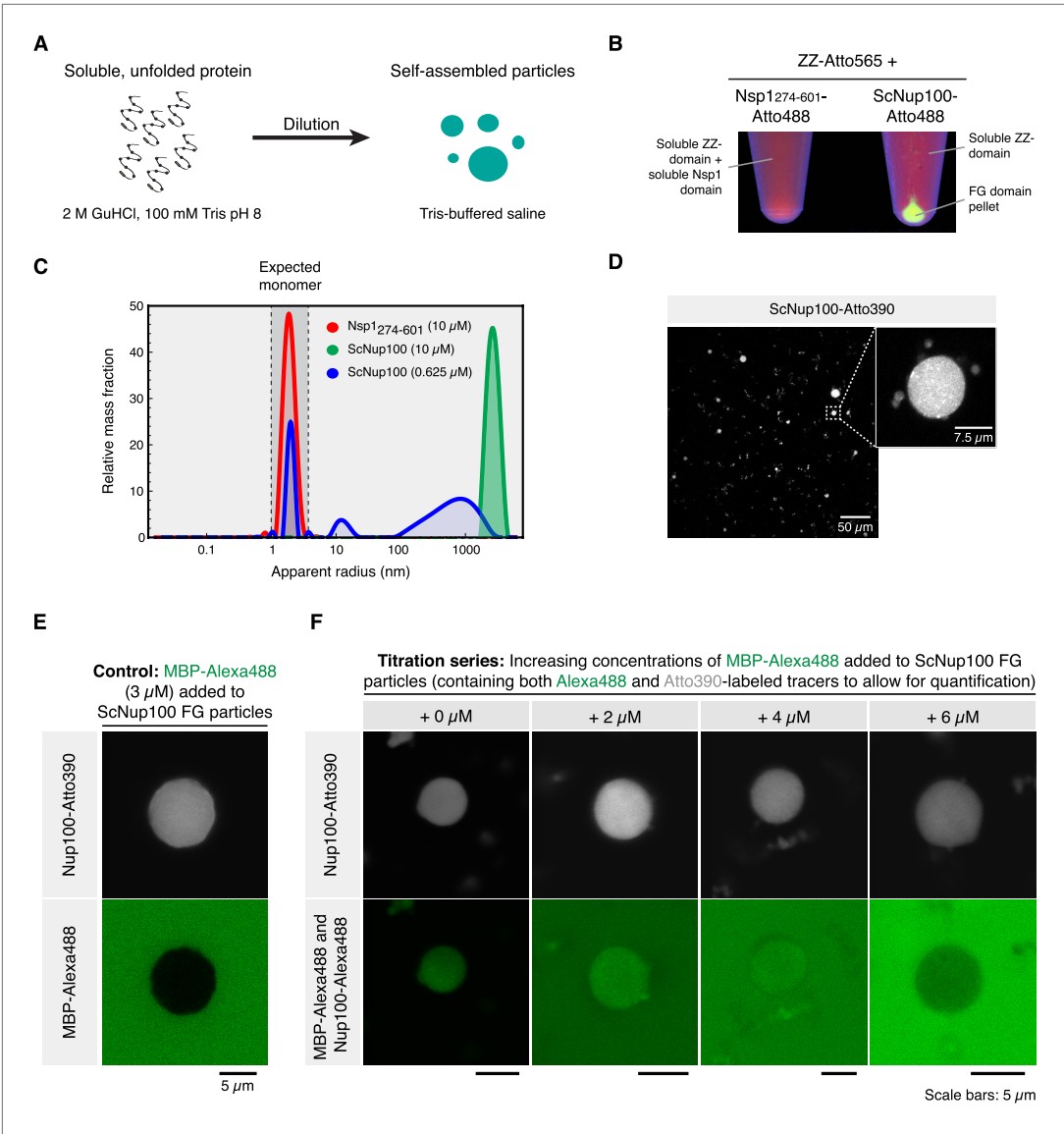

**Figure 1**. Dilute Nup100 FG domain solutions spontaneously undergo phase-separation. (**A**) Illustration of the experimental design. (**B**) Cohesive FG domains self-assemble into FG phases that can be collected by centrifugation. Two stock protein solutions were prepared in 2 M guanidinium hydrochloride (GuHCl), 100 mM Tris/HCl pH 8.0. They contained 300 µM of a Z-domain tandem fusion (labelled 1:1 with Atto565 maleimide) and 300 µM FG domain from either or Nsp1 (residues 274–601) or Nup100. 5% of the FG domain molecules carried an Atto488 maleimide label. 16.7 µl of each solution was diluted with 500 µl 50 mM Tris/HCl pH 7.5, 150 mM NaCl (TBS). Photographs show test tubes after ultracentrifugation (100,000×$g$, 30 min), illuminated at 366 nm. Note that the Nup100 FG domain pelleted, while the non-cohesive Nsp1 FG repeats and the globular ZZ-domain remained soluble. (**C**) FG particle formation at different concentrations. Label-free ScNsp1$_{276-601}$ and ScNup100 FG domains were diluted from 400 µM stocks (in 2 M GuHCl) to the indicated concentrations with TBS. Formed particles were analysed by Dynamic light scattering (DLS) using a DynaPro NanoStar instrument (Wyatt Technologies). Two data sets, comprising each 100 acquisitions à 5 s, were averaged for each dilution. The Dynamics 7.1.5 software was used for autocorrelation analysis and computation of size distributions. (**D**) Confocal laser-scanning microscopy (CLSM) images showing an overview and zoom-in of ScNup100 FG particles. (**E**) FG particles exclude inert molecules. Particles were formed with 10 µM ScNup100 FG domain and 0.5 µM Atto390-tracer. Particles were mixed with Alexa488-labeled maltose binding protein (MBP), which remained excluded from the particle and thus qualified as an internal standard for Alexa488 fluorescence. (**F**) Estimation of FG domain concentration within FG particles. Particles were formed with 10 µM unlabelled, 0.5 µM Atto390- and 14 nM Alexa488-labelled ScNup100 FG-domain. CLSM images were taken after adding different dilutions of MBP-Alexa488, which served as an internal fluorescence standard. Correlating extra-particle Alexa488 signals (originating from MBP) with the known supplied MBP concentrations and matching them with the intra-particle Alexa488 signals (originating from 1/715th of the FG-domain molecules) suggests that an average particle contains ≈4.5 mM (≈275 mg/ml) FG-domain. This corresponds to ≈200 mM FG motifs.

As an initial example, we chose the FG domain of *S. cerevisiae* Nup100, because it is a well-studied Nup98 homolog that can restore a functional barrier in FG Nup-depleted *Xenopus* NPCs (*Hülsmann et al., 2012*), but it has not yet been tested if this domain is also sufficient for assembling a selective hydrogel in vitro. In the first series of experiments, we prepared a 1000 µM Nup100 FG domain stock solution, supplemented by 2 M guanidinium hydrochloride in order to initially keep the domain molecules in a non-interacting state. The solution was then rapidly diluted in 100 volumes of a neutral Tris/NaCl buffer, which lowered the protein concentration to 10 µM and the guanidinium ion concentration to negligible levels (*Figure 1A*). Remarkably, the Nup100 FG domain solution turned instantaneously turbid, pointing to a very rapid phase-separation and formation of small particles or liquid droplets, which can be easily recovered by centrifugation (*Figure 1B*). We tried to record a time course of this reaction (by static light scattering) but had to realise that the reaction had already reached its endpoint before we could place samples into the instrument and start the measurement (after 10–30 s).

When applying dynamic light scattering (DLS) to the 10 µM Nup100 FG domain sample, we observed a very prominent particle population with diameters ranging mostly between 2 and 8 µm (*Figure 1C*). At a lower concentration (0.625 µM), we observed a broader main peak with 0.4–4 µm particles (accounting for ≈60% by mass), another peak with 25 nm assembly intermediates (10%) and residual monomers (30%). The remaining monomer concentration can be taken as an estimate for the critical concentration for this phase-separation, which hence should be in the range of ≈200 nM FG domain or ≈10 µM FG repeat units. Yet, already the anchorage of FG domains to the NPC scaffold should result in a ≥200 times higher local concentration (2 mM repeat units; see above). This ≥200-fold oversaturation should thus drive FG phase-separation to completion also in real NPCs. The rapid onset of the in vitro FG particle formation further supports this view, as it indicates that self-association of Nup100 FG domains is not impeded by a kinetic hurdle.

As a control, we analysed the regular and highly charged part of the *S. cerevisiae* Nsp1p FG domain (residues 274–601) and observed that it failed to form particles at 10 µM domain concentration (*Figure 1B,C*). This confirms the earlier observations that this FG subdomain is of low cohesiveness (*Ader et al., 2010*; *Yamada et al., 2010*) and fails to assemble a functional barrier in *Xenopus* NPCs that lack the Nup98 FG domain (*Hülsmann et al., 2012*).

Confocal laser-scanning microscopy (CLSM) of Nup100 FG particles, formed in the presence of 5% (vol/vol) Atto390-labeled FG domain tracers, indicated that the particle size increased with higher initial concentration (not shown). Most particles were of nearly spherical shape, whereby the evident deviations from perfect spheres indicated that the phase-separated objects represent solids rather than liquids (*Figure 1D*; further evidence for this assumption is provided below).

As a next step, we wanted to determine the Nup100 FG domain concentration in the formed particles. For that we recorded the local fluorescent intensity of the added Alexa488 FG domain tracer, determined absolute fluorophore concentrations by calibration with a series of internal Alexa488-MBP standards (*Figure 1E,F*), and used this number to derive an average intra-particle concentration of the Nup100 FG domain of 275 mg/ml (corresponding to 4.7 mM FG domain or 200 mM FG motifs). Further assuming a specific partial volume for the protein part of 0.73 ml/g, one can estimate that the polypeptide accounts for 20% and water (respectively buffer) for the remaining 80% of the particle volume. These numbers are well in line with the assumption that these particles indeed represent hydrogels.

## ScNup100 FG particles are highly selective FG phases

We subsequently used CLSM to assess how fluorescent permeation probes would partition into the self-assembled Nup100 FG particles. As shown in *Figure 2A*, the particles allowed for a strong accumulation of probes with NTR-like properties. This applied to NTF2, Importin β complexed with an Atto488-labelled IBB-domain or an IBB-GFP fusion, or even a 520 kDa octameric complex comprising four copies of Importin β and the IBB-ZsGreen homotetramer. The particles were imaged after just ≈3 min of NTR influx. Nevertheless, particle:buffer partition coefficients of ≈200 were observed, and the NTRs showed already a rather even distribution between the centres and peripheries of the 5–10 µm-sized particles. A comparison with computer simulations (*Figure 3*) suggests that the influx was so rapid that diffusion within the surrounding buffer, but not the actual particle-entry, must have become rate-limiting. The intra-particle NTR distributions are consistent with intra-particle diffusion coefficients of ≥0.1 µm²/s, implying that it would take these transported species no more than

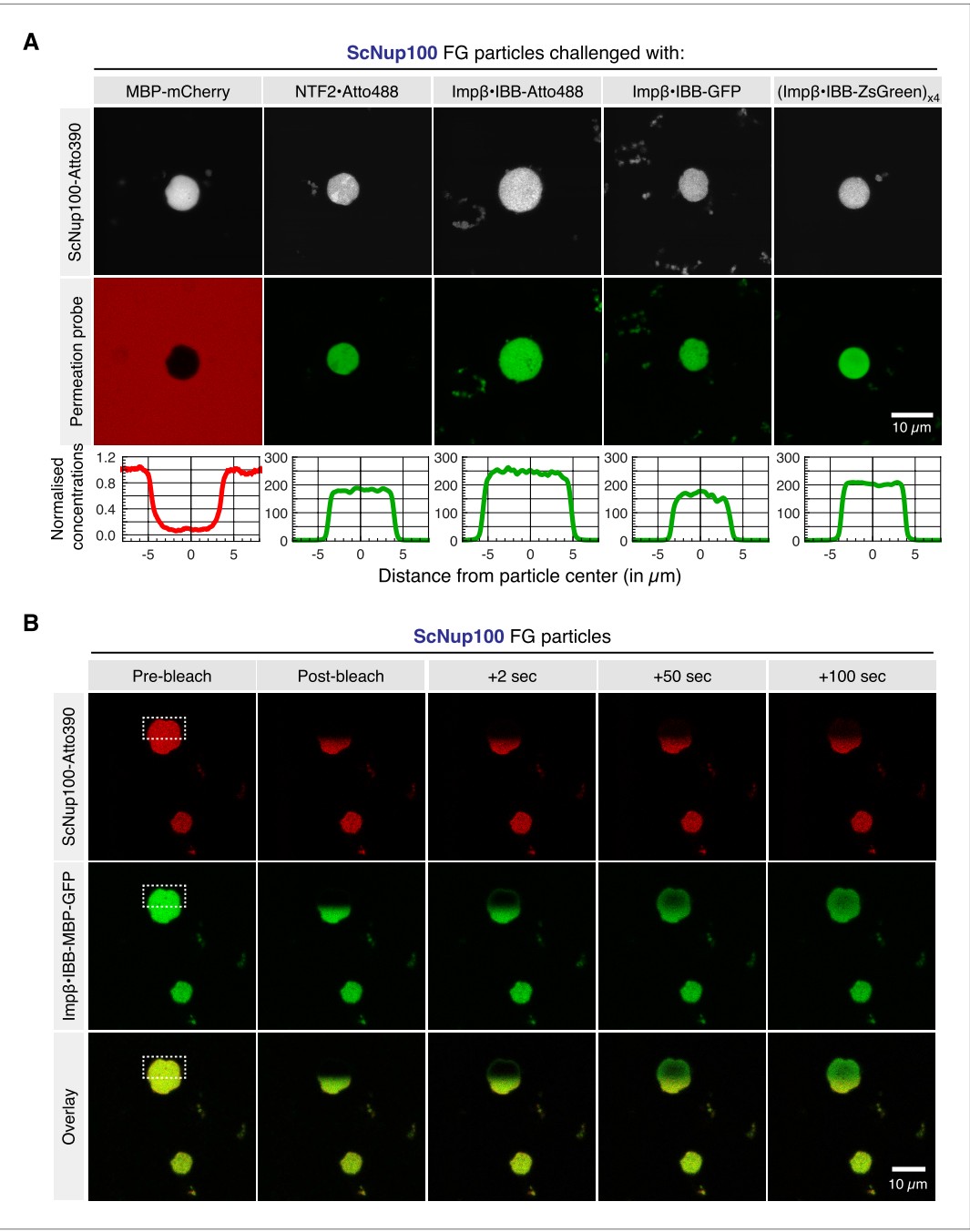

**Figure 2**. Permeability properties of ScNup100 FG particles. (**A**) FG particles were formed at 10 µM ScNup100 FG domain concentration (including 5% Atto390-tracer), as described above. 3 µM of the passive permeation probe MBP-mCherry or 1 µM of the indicated active permeation species were added (concentration referring to substrate monomers). CLSM images were taken ≈2–3 min later, using the 405 nm, 488 nm, or 561 nm laser lines for exciting the FG tracer, active or passive permeation probes, respectively. (**B**) Intra-FG particle dynamics of FG domains and NTR·cargo complexes. ScNup100 FG particles were formed as in *Figure 1E* and challenged with 1 µM of a yeast Impβ•IBB-MBP-GFP complex. CLSM images show two particles. One of them was photobleached at 405 and 488 nm in one hemisphere. Fluorescence recovery of the FG domain tracer as well as of the NTR·cargo complex was detected over time.

10 milliseconds to traverse a 40-nm thick NPC barrier made of the same material. This agrees well with previously observed NPC dwell times of NTF2 or importin β during a successful pore passage (*Yang et al., 2004*; *Kubitscheck et al., 2005*; *Yang and Musser, 2006*).

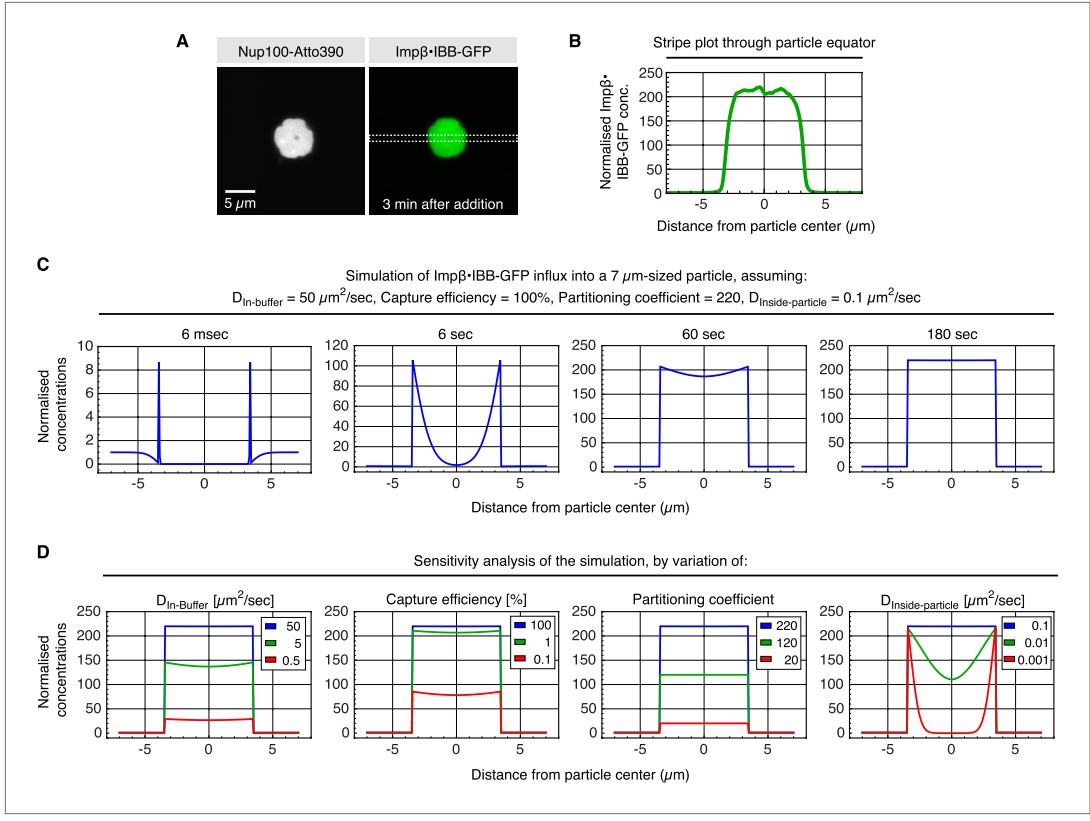

**Figure 3**. Estimate for kinetic parameters for Importinβ•IBB-GFP influx into ScNup100 FG particles. Influx of the NTR·cargo complex into the FG particles had to be performed in suspension, which implied that we had to wait until particles had settled to the bottom of the slide (≈3 min) before concentration profiles across a particle and surrounding buffer could be recorded. Then, however, the endpoint of accumulation was essentially reached already. In order to nevertheless estimate kinetic parameters for particle-entry, we simulated the process and asked which parameter set would be consistent with the observed cargo distribution at the 3 min time point. These parameters included the partition coefficient (220) and 7 µm particle diameter (both measured directly), the diffusion coefficient in buffer ($D_{buffer}$ = 50 µm²/s, derived by the Stokes–Einstein equation from the radius of the Impβ•IBB-GFP complex and the viscosity of the buffer), as well as the intra-particle diffusion coefficient ($D_{Particle}$ = 0.1 µm²/s), which was the smallest that allowed an even intra-particle distribution of the cargo at the 3 min timepoint. 'Capture efficiency' describes the probability that a colliding NTR·cargo complex gets captured by the particle. Simulations were performed in Mathematica 9.0 and exploited the spherical symmetry of the particle to simplify the system of differential equations (see ***Supplementary file 1*** for the Mathematica code and more detailed explanations). (**A**) ScNup100 FG particles were formed at 10 µM and 30 min later challenged with 1 µM Impβ•IBB-GFP complex. CLSM image was taken after another 3 min. (**B**) Impβ•IBB-GFP concentration profile across the area indicated in panel **A**. Signal was normalized to the concentration in buffer. (**C**) Simulation of influx for indicated parameters and time points. (**D**) Sensitivity analysis, varying the parameters used in **C**. It revealed that diffusion in buffer, the partition coefficient, and diffusion inside the particle, but not the capture efficiency, are limiting for the influx process.

At the same time, we observed that the inert MBP-mCherry fusion (65 kDa) remained well excluded from the particles' interior (see ***Figure 2A***). The respective partition coefficients of <0.1 were far lower than the estimated fraction of the particles' volume that is occupied by the solvent, which in turn is consistent with the assumption of a sieve structure causing a size-exclusion effect.

## Though FG particles are solids, NTRs traverse them like a liquid

The phase-separation of cohesive FG domains from dilute aqueous solutions draws an interesting parallel to RNA granules, which are membrane-free compartments formed by multivalent interactions between RNAs and RNA-binding proteins. RNA granules are also of near-spherical shape and this shape has been attributed to a liquid state and surface tension effects (***Brangwynne et al., 2009***).

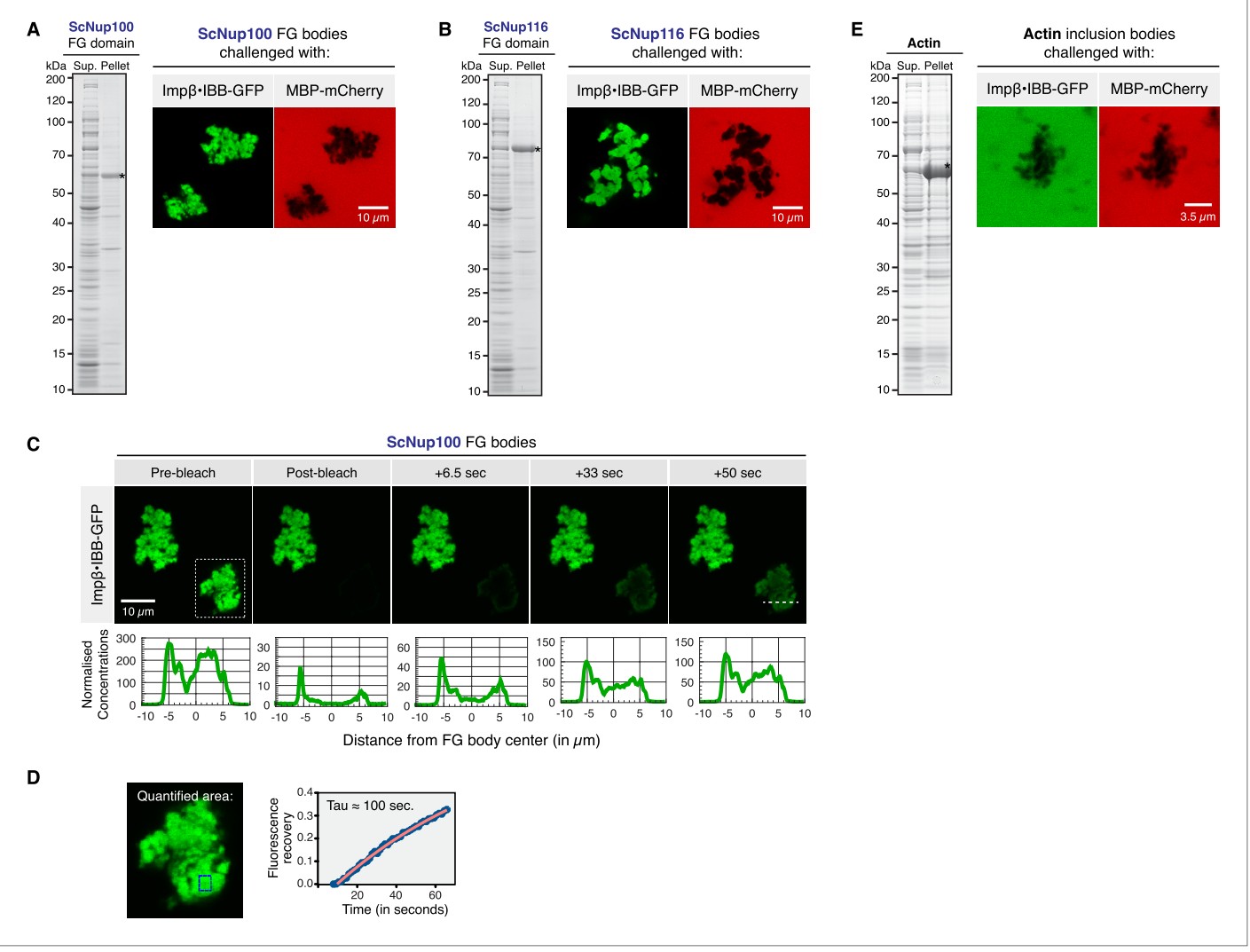

**Figure 4**. In vivo assembly of selective FG phases. For the undistorted in vivo formation of FG bodies, the ScNup100 and ScNup116 FG domains were expressed in *Escherichia coli* (NEB Express) following induction with 0.5 mM IPTG at 30°C. Cells were pelleted, resuspended in TBS, and lysed with 1 mg/ml lysozyme, 20 µg/ml DNAse I and 0.5% Tween 20. Insoluble 'FG bodies' were recovered by centrifugation at 10,000×*g*. SDS-PAGE analysis of supernatant ('Sup') and pellet fractions and the permeability properties of the recovered ScNup100 (**A**) and ScNup116 (**B**) FG bodies are shown. Crude FG bodies were washed twice in TBS and their permselectivity analysed as described in *Figure 2*. (**C**) Mobility of NTR·cargo complexes in FG bodies. ScNup100 FG bodies were challenged with 1 µM Impβ•IBB-GFP and photobleached, and fluorescence recovery was detected over time. Note that the NTR·cargo complex was mobile also within the in vivo formed FG bodies. The white dashed line in the 50 s frame indicates the region analysed for the line plots shown below the images. (**D**) Fluorescence recovery over time in the area indicated by the blue box in the zoom-in of the particle outlined by the dashed lines in **C**. The analysed region lies approximately 1 µm inside of the particle. Fluorescence recovery occurred with a time constant of ≈100 s. Note that this involved not only intra-particle diffusion, but also the uptake from the buffer against a ≈200-fold concentration gradient. (**E**) In contrast to the FG bodies, actin inclusion bodies did not enrich the Impβ•IBB-GFP species, but in fact excluded it like MBP-mCherry.

Given these similarities, we wanted to probe the physical conditions of the obtained FG phases and used FRAP for this purpose (*Figure 2B*). After bleaching the Atto390 fluorophore of labelled FG domains, we observed that bleached patterns within the Nup100 FG particles remained stable for some minutes at least. This indicates a solid state of the particles and stable interactions between FG domains. The simultaneously bleached NTR-signal was, however, far more dynamic. The particles' peripheries showed a clear signal recovery already within two seconds, indicating a very rapid exchange with the surrounding medium. This implies that this NTR-species can rapidly exit the particle—despite its very high partition coefficient. The fluorescent signal in the centre of a particle

with a radius of ≈4 μm recovered with a half time of ≈40 s. This clearly shows that NTRs are actually very mobile within a rather static FG phase.

## Formation of selective FG phases within living cells

We had previously prepared FG hydrogels by quickly dissolving the TFA (trifluoroacetic acid) salt of a lyophilised FG domain to ≈200 mg/ml, and one could argue that gels formed only because the domains were forced artificially to such a high local concentration. We now demonstrated that initially very dilute aqueous solutions of Nup98 FG domains phase-separate into an FG-rich phase (see *Figure 1* and below). However, in order to suppress intermolecular contacts prior to this dilution step, we had to keep the concentrated protein solution in ≥2 M guanidinium hydrochloride. One could therefore argue that such transient denaturation is still non-physiological and might have caused an artificial FG hydrogel formation that otherwise would not have occurred.

In order to address this issue, we avoided any denaturing treatment during the next steps. Specifically, we tested whether Nup98-derived FG domains also self-assemble into selective phases, when they are simply expressed in the bacterium *Escherichia coli* and thus are not exposed to any potentially structure-changing manipulation. Following recombinant expression (after modest induction), we resuspended the bacteria in a physiological buffer, gently disrupted the cells by lysozyme treatment, and subjected the lysate to a 10,000×*g* centrifugation step. We observed that the FG domains from Nup100 and Nup116 (the second *S.c.* Nup98 paralog) pelleted under these conditions (*Figure 4A,B*). Given that the centrifugation was performed with only a k-factor of ≈1000 S, it thus appeared that these FG domains had formed rather large structures.

At first glance, the insoluble material resembled ordinary inclusion bodies that result when 'difficult-to-fold-proteins' (for example recombinant actin; *Figure 4E*) form irreversible aggregates during overexpression in a prokaryotic host. One critical difference is, however, that the initially insoluble 'FG bodies' slowly dissolved when soluble FG domain molecules were removed from the equilibrium, for example during repeated steps of pelleting and resuspending in fresh aliquots of buffer (not shown).

Microscopic analysis revealed that the Nup100 and Nup116 'FG bodies' were 10 μm-sized irregularly shaped particles that probably had been 'glued' together from smaller species during the centrifugation steps. They nearly perfectly excluded our passive permeation marker MBP-Cherry and yet allowed a very efficient intra-particle accumulation of the NTR·cargo complex Impβ·IBB-GFP, reaching again a particle:buffer partition coefficient of around 200 (*Figure 4A,B*).

In *Figure 4C*, we bleached the Impβ·IBB-GFP signal in an area that comprised an entire Nup100 FG body. Fluorescence recovery occurred with a time constant of ≈100 s in the interior of the particles (*Figure 4D*). This is remarkably fast, considering that (i) recovery could occur only through the influx from the buffer, (ii) that the concentration in the buffer was just 0.5% of that inside the particles, and (iii) that the dimension of the particle was almost 200-fold larger than the presumed NPC barrier, i.e. when scaled down to NPC dimensions, recovery would have occurred with a time constant of a few milliseconds.

Earlier studies already reported that overexpressing, for example, a YFP-Nup100 FG repeat fusion in *S. cerevisiae* (*Patel et al., 2007*) or GFP-fused to the human Nup98 FG domain in HeLa cells (*Xu and Powers, 2013*), can result in characteristic intra-cellular foci. Yet, the composition, dependence on host factors and barrier properties of such assemblies had not been evaluated. *Figure 4* now demonstrates that in vivo formed minimalistic Nup98 FG phases feature a striking NPC-like permselectivity. It also shows that phase-separation requires neither accessory eukaryotic factors nor sophisticated experimental manipulations. Instead, the assembly of FG bodies relies exclusively on the strong intrinsic propensity of these FG domains to interact.

## Nup98 FG domains from diverse species share the propensity to form highly selective FG particles

If self-assembly of barrier-critical FG domains into selective FG phases is fundamental for NPC function, then one can expect this phenomenon to be conserved throughout the eukaryotic tree of life. We decided to test this assumption by comparing ten different Nup98 FG domains from nine divergent species, namely: Nup98 from human (*Radu et al., 1995*), *Branchiostoma floridae* (representing lancelets), *Drosophila melanogaster* (representing insects; *Presgraves et al., 2003*), NPP-10/Nup98 from *Caenorhabditis elegans* (representing nematodes; *Voronina and Seydoux, 2010*), the two

already mentioned paralogs Nup100 and 116 from *Saccharomyces cerevisiae* (representing fungi; *Wente et al., 1992*), *Dictyostelium discoideum* Nup220 (representing amoebas), *Arabidopsis thaliana* Nup98B (representing plants; *Tamura et al., 2010*), the macronuclear MacNup98A from *Tetrahymena thermophila* (representing ciliates; *Iwamoto et al., 2009*) as well as Nup158 from *Trypanosoma brucei* (representing euglenozoans/excavates; *DeGrasse et al., 2009*). Criteria for this selection had been a wide sampling of species and sequence diversity, a preference for well-studied model organisms, and exclusion of FG domains shorter than 400 residues. The resulting selection covered all major eukaryotic clades (See *Figure 5A*), with the exception of Rhizaria, where genome analysis has lagged

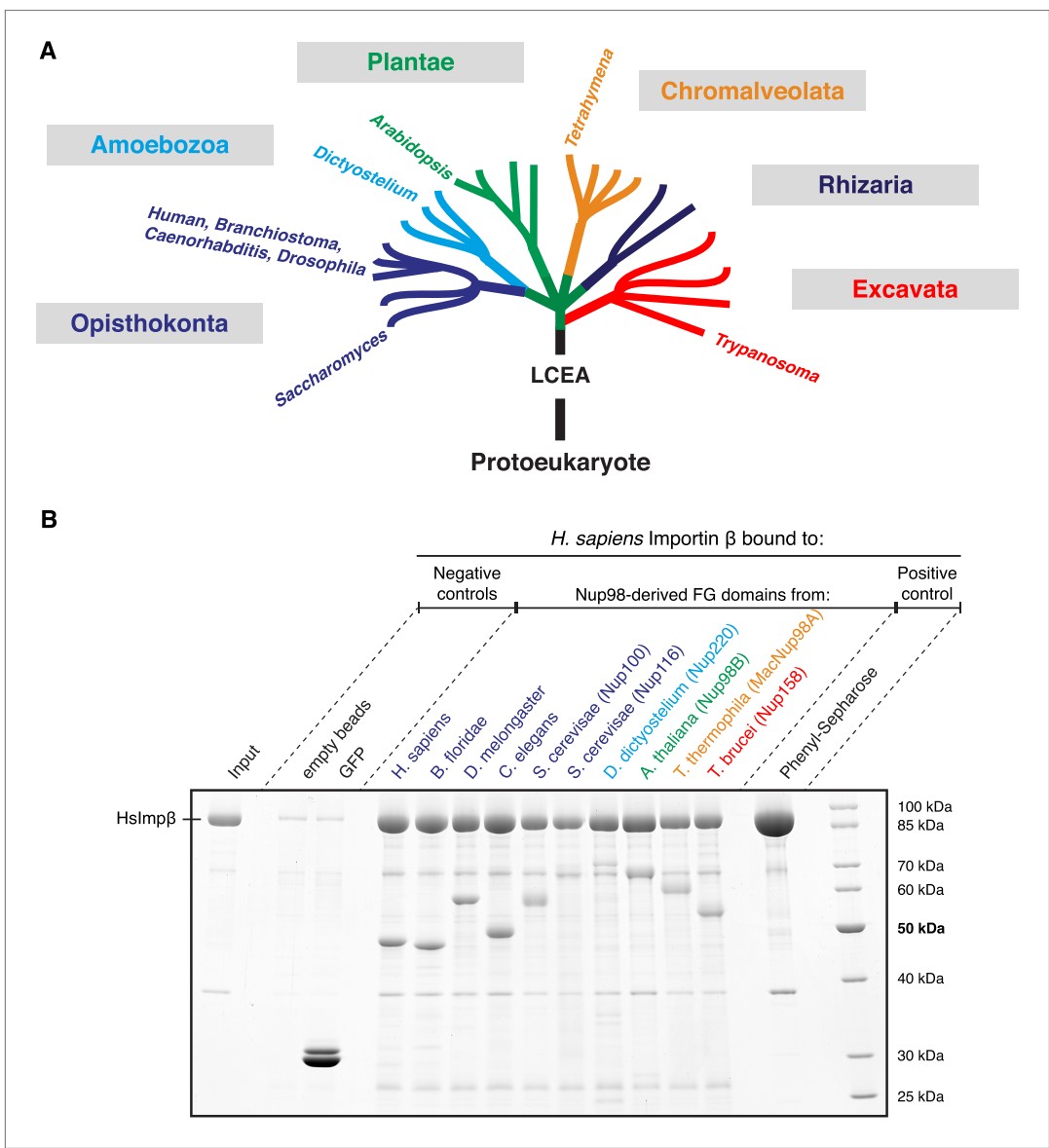

**Figure 5**. Nup98 FG domains of diverse evolutionary origin bind human Importin β. (**A**) This study analyses ten Nup98 FG domains from nine species. Cartoon illustrates their positions within the eukaryotic tree of life (adapted from *Keeling et al., 2005*). See *Supplementary file 2* for complete sequences. LCEA denotes the position of the last common ancestor to all eukaryotes. (**B**) Indicated His-tagged FG domains (30 μg each) were immobilized on 30 μl PEG-passivated Ni(ii) chelate beads and rotated for 2 hr at 4°C with 1 μM untagged human Importin β (400 μl). Bound prey and immobilized baits were co-eluted with SDS/Imidazol and analysed by SDS-PAGE/Coomassie-staining. Note the incomplete elution of the *Saccharomyces* Nup116 and *Dictyostelium* Nup220 FG domains. Phenyl-Sepharose served as a positive control (*Ribbeck and Görlich, 2002*). Binding was in 25 mM Tris/HCl pH 7.5, 100 mM NaCl, 1 mM MgCl$_2$, 0.5% PEG4000, 5 mM DTT.

far behind and only Nup98s with apparently short FG domains were listed in databases at the time of our analysis.

As a first step, we constructed bacterial expression vectors, and recombinantly expressed, purified, and immobilized all ten Nup98 domains. We observed that all of them bound human Importin β specifically and to a similar extend (*Figure 5B*). This suggests that the mode of FG domain·NTR-interaction has not changed dramatically during eukaryotic evolution.

We next extended our analysis of spontaneous FG phase formation to the entire set of Nup98 FG domains. We observed that all of them showed the same readiness to phase-separate and form FG particles as the ScNup100 FG domain (*Figures 6 and 7*). The size distribution of particles differed between the various FG domains, which perhaps reflects different particle seeding and growth rates. But otherwise, the particles were mostly spherical and showed a rather even intra-particle FG domain distribution. In this species comparison, they were also remarkably similar in their intra-particle FG domain concentration (average 250 mg/ml, range 175–350 mg/ml; see *Table 1*). Furthermore, not in a single case did the critical FG domain concentration for the phase-separation exceed 1 µM or 50 µg/ml, even when FG domains were O-glycosylated (*Table 1*; see also *Labokha et al., 2013*). Remember that the local Nup98 FG domain concentration at NPCs should be at least 50 times higher, even if one assumes that the domains initially do not interact. Thus, a phase-separation of Nup98 FG domains into selective FG hydro-gels should be thermodynamically highly favoured for the stoichiometry settings of authentic NPCs. Moreover, the striking conservation of the underlying biophysical properties throughout the eukaryotic kingdom can be taken as a very strong argument for a fundamental functional relevance.

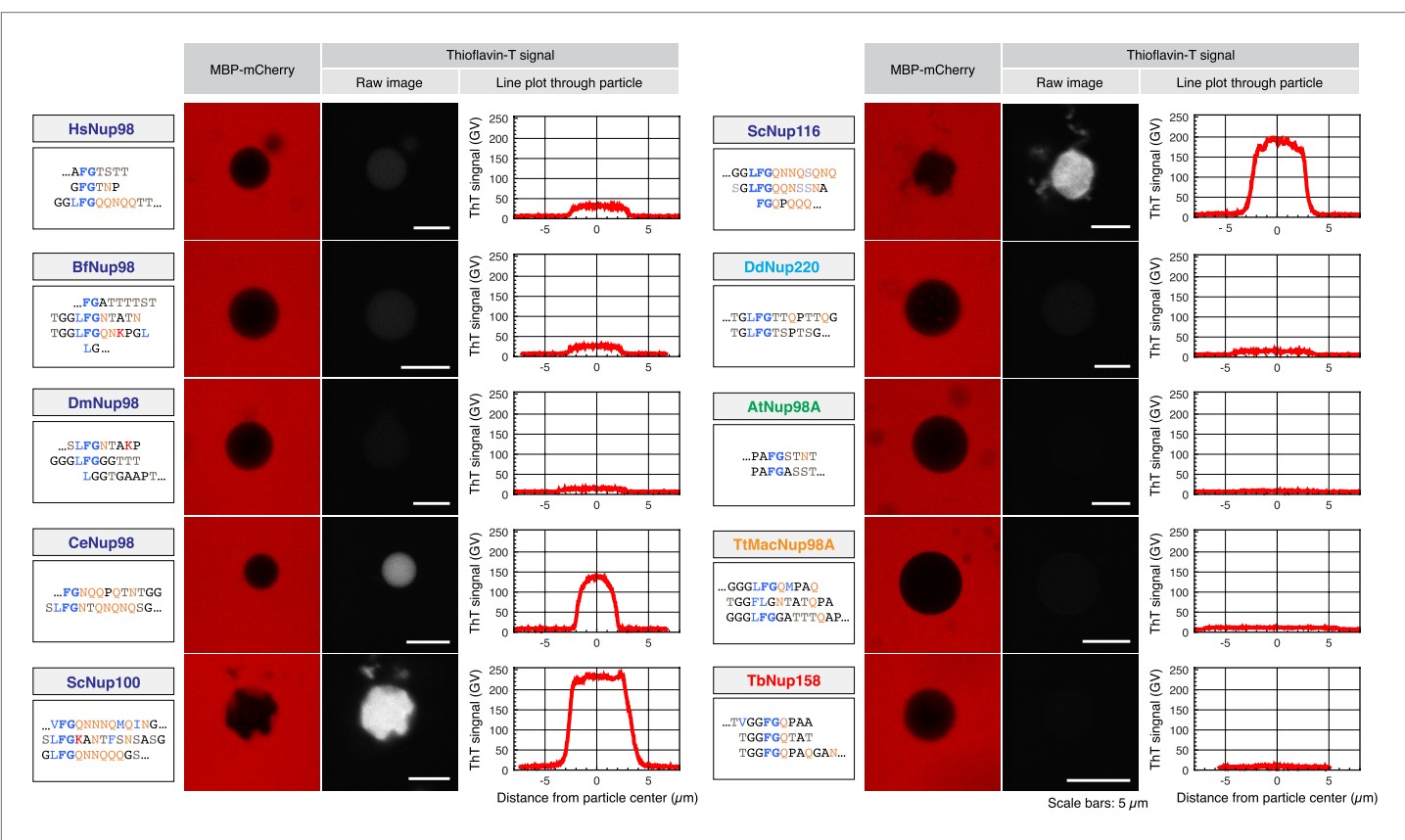

**Figure 6**. Different modes of inter-FG repeat interactions within FG particles. The indicated Nup98 FG domains were studied. Representative repeat sequences are shown in single letter code (see *Supplementary file 2* for complete sequences). Particles were formed at 5 µM (HsNup98, TtMacNup98, TbNup158) or 10 µM FG domain concentration (all other FG domains). The suspensions were afterwards supplemented with 3 µM MBP-mCherry and 1 µM ThioflavinT, a diagnostic dye for the presence of amyloid-like cross-β-structures. Particles were detected based on exclusion of MBP-mCherry. ThioflavinT was excited at 405 nm and detected in a 460–500 nm window. Graphs show quantitations for the measured signals (gray value scales). ScNup100 FG particles gave the strongest signal. For *Table 1*, all Thioflavin signals were normalized to the Nup100 signal.

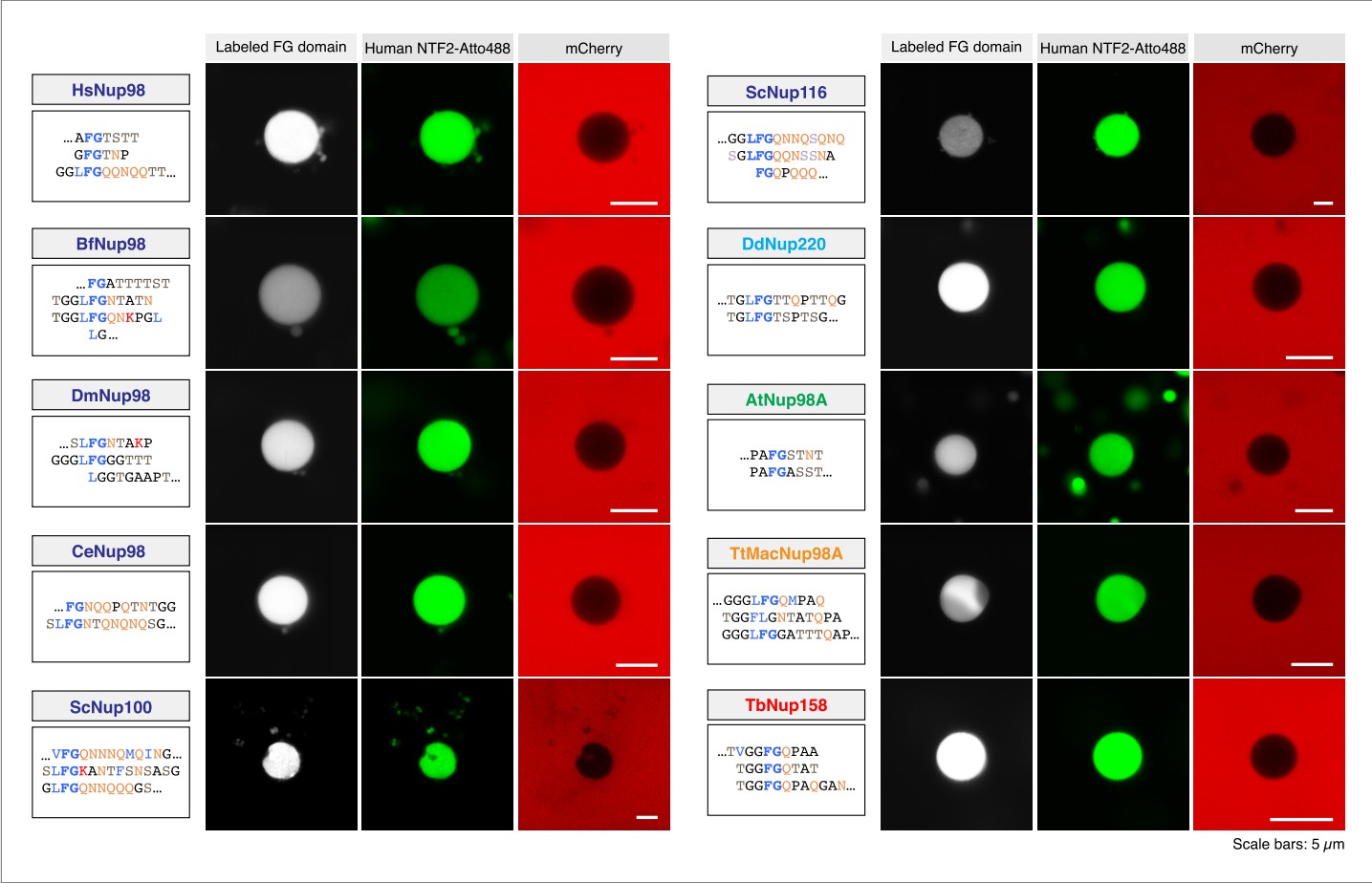

**Figure 7**. Nup98 FG domains from distant eukaryotic clades self-assemble into highly selective FG particles. Particles were formed as described in *Figure 6*, followed by the addition of either 1 μM NTF2-Atto488 or Impβ•IBB-GFP and 3 μM mCherry as active and passive permeation probes, respectively. The experimental setup was as in *Figure 2A*.

## Nup98 FG phases may or may not rely on cross-β structures

As expected from the extreme evolutionary distances separating the host species, the sequences of the ten analysed Nup98 FG domains are actually rather diverse. They differ, for example, quite widely in their NQ-content (*Table 1*; *Supplementary file 1*), i.e., in a parameter that has been linked to the formation of amyloid-like structures, also in NQ-rich hydrogels (*Alberti et al., 2009*; *Ader et al., 2010*). To test if these sequence differences also translate into different contents of amyloid-like cross-β sheets, we stained the particles with ThioflavinT—a compound whose 480 nm fluorescence emission is enhanced upon binding to NQ-rich and other amyloid-like cross-β structures (*Vassar and Culling, 1959*; *Khurana et al., 2005*). Particles derived from the most NQ-rich FG domains, Nup100 (28%) and Nup116 (26%) showed indeed by far the strongest ThioflavinT signal, supporting that cross-β structures can occur not only in amyloid fibres, but also in self-assembled FG particles (*Figure 6* and *Table 1*). The NQ content alone is, however, not necessarily a reliable predictor for the presence of cross-β sheets. The *C. elegans* Nup98 and the *Tetrahymena* MacNup98A FG domains, for example, have a very similar NQ-fraction (18%), yet we observed a clear ThioflavinT signal only for *C. elegans* FG particles. The *Tetrahymena* MacNup98A FG particles were essentially ThioflavinT-negative, perhaps because their high glycine content counteracts β-sheet formation and/or stability. FG particles from the other species also showed at most a weak ThioflavinT signature. Taken together, this suggests that particles from the selected FG domains sample different modes of inter-FG repeat interactions, and we were curious to find out if this would translate into different permselectivities.

**Table 1.** Key descriptors of FG particle constitution

| | Molecular weight* | Number of residues* | Number of FG motifs* | Estimated critical concentration† | | Estimated intra-particle concentration‡ | NQ-content* | Relative Thioflavin-T signal§ |
|---|---|---|---|---|---|---|---|---|
| | | | | FG domain | FG motifs | | | |
| *Homo sapiens* HsNup98 | ≈49 kDa | 500 | 39 | ≈25 nM ≈1 µg/ml | ≈1 µM | ≈175 mg/ml | 11% | 11% |
| *Branchiostoma floridae* BfNup98 | ≈46 kDa | 479 | 40 | ≈20 nM ≈1 µg/ml | ≈1 µM | ≈200 mg/ml | 8% | 8% |
| *Branchiostoma floridae* BfNup98 + GlcNAc | ≈55 kDa | 479 | 40 | ≈150 nM ≈10 µg/ml | ≈6 µM | ND | 8% | ND# |
| *Drosophila melongaster* DmNup98 | ≈56 kDa | 581 | 46 | ≈200 nM ≈10 µg/ml | ≈9 µM | ≈300 mg/ml | 10% | 3% |
| *Caenorhabditis elegans* CeNup98 | ≈48 kDa | 494 | 36 | ≈700 nM ≈40 µg/ml | ≈25 µM | ≈300 mg/ml | 18% | 52% |
| *Saccharomyces cerevisiae* ScNup100 | ≈58 kDa | 578 | 43 | ≈175 nM ≈10 µg/ml | ≈7.5 µM | ≈275 mg/ml | 28% | 100% |
| *Saccharomyces cerevisiae* ScNup116 | ≈65 kDa | 737 | 47 | ≈700 nM ≈50 µg/ml | ≈33 µM | ≈350 mg/ml | 26% | 78% |
| *Dictyostelium discoideum* DdNup220 | ≈68 kDa | 719 | 56 | ≈125 nM ≈10 µg/ml | ≈7 µM | ≈300 mg/ml | 12% | 4% |
| *Arabidopsis thaliana* AtNup98B | ≈66 kDa | 668 | 52 | ≈25 nM ≈1 µg/ml | ≈1.5 µM | ≈200 mg/ml | 13% | 1% |
| *Tetrahymena thermophila* TtMacNup98A | ≈61 kDa | 666 | 42 | ≈25 nM ≈1 µg/ml | ≈1 µM | ≈175 mg/ml | 18% | 2% |
| *Trypanosoma brucei* TbNup158 | ≈50 kDa | 565 | 58 | ≈300 nM ≈15 µg/ml | ≈17.5 µM | ≈250 mg/ml | 12% | 2% |

*All values are given for full-length FG domains (including GLEBS domains). The molecular weight of glycosylated BfNup98 was estimated by SDS-PAGE.

†The critical concentrations for phase separation were estimated as described in *Figure 1C* and the methods section.

‡Intra-particle FG domain concentrations were estimated as described in *Figure 1F* and the methods section.

§Thioflavin-T signals are normalised to the ScNup100 Thioflavin-T signal; also see *Figure 6*.

#Experiments with macroscopic hydrogels suggest that the Thioflavin-T signal of glycosylated BfNup98 FG particles is even lower than the observed signal for the non-glycosylated BfNup98 FG particles.

## Conserved permselectivity of Nup98 FG phases from diverse species

Strikingly, we observed that Nup98 FG particles from all ten Nup98 FG domains efficiently excluded not only the still rather large MBP-mCherry fusion (≈75 kDa) (*Figure 6*) but also rejected the far smaller mCherry (≈25 kDa) (*Figure 7*). At the same time, these FG phases allowed a high or very high accumulation of NTF2 (*Figure 7*). In each case, the partition coefficient of NTF2 was at least 1000 times higher than that of mCherry.

A FRAP analysis revealed that the solid state of the Nup100 FG particles and low mobility of the phase-separated Nup100 FG domain were no exception but also applied to other FG particles, for example, formed by the evolutionary very distant *Tetrahymena* MacNup98A FG domain (*Figure 8*). Our analysis also revealed an extremely rapid exchange of NTF2 between the TtMacNup98A particles and the surrounding buffer, as well as a very high mobility of NTF2 inside these particles.

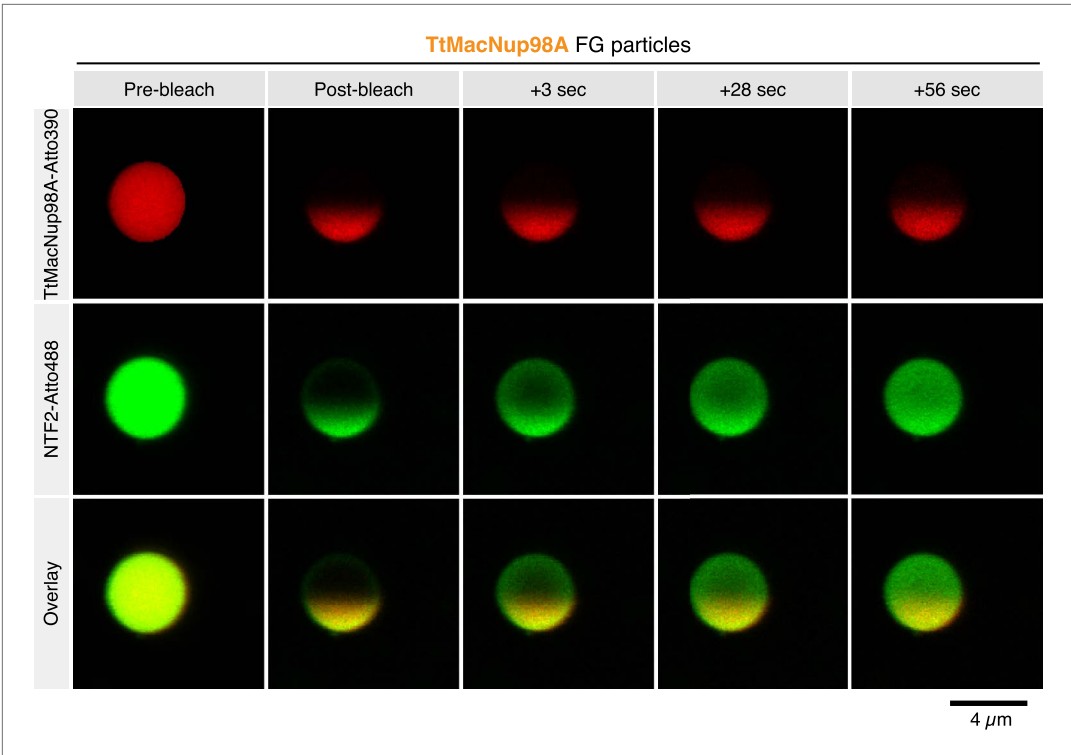

**Figure 8**. Intra-FG particle dynamics of FG domains and NTRs. TtMacNup98A FG particles were formed with 5 µM unlabeled and 25 nM Atto390-labeled TtMacNup98A FG domain and challenged with 1 µM NTF2-Atto488. 5 min after NTR addition, particles were photobleached at 405 and 488 nm in one hemisphere, and fluorescence recovery of the tracer and NTR followed over time.

## Sequence features of Nup98 FG domains

We observed that aqueous Nup98 FG domain solutions spontaneously separate into FG phases with just the 'right' FG domain concentration and structure for creating a barrier with NPC-typical permselectivity. This held true for Nup98 FG domains from nine evolutionary very distant species—a clear indication for a strong evolutionary pressure to maintain the underlying physicochemical properties. As an additional indication, we noticed extreme sequence conservation amongst vertebrates, covering ≈400 million years of evolution. The Nup98 FG domains from human and the fish *Lepisosteus oculatus*, for example, share ≈70% identical residues, which is close to the ≈75% identities between the corresponding globular autoproteolytic Nucleoporin 2 domains (*Figure 9*). Moreover, most of the observed exchanges were just conservative permutations between T, S, A, and N within the spacers, while spacer lengths and the type of FG motif at a given position remained extremely conserved. The Nup98 FG domain is thus an exception from the rule that intrinsically disordered domains change rapidly during evolution (*Denning and Rexach, 2007*). In fact, this indicates that the Nup98 FG domain engages in critical interactions along its entire sequence and that deviations from this optimal sequence are not well tolerated.

With greater evolutionary distance, differences in the preferred FG sequence context and inter-FG spacer compositions became evident. Ciliates, for example, prefer GLFG motifs, fungi SLFG or GLFG, plants PFG, PAFG, or xFG and Trypanosomes GGFGQ motifs (*Table 2*). Likewise, fungi prefer very NQ-rich inter-FG spacers, lancelets very T-rich, and trypanosomes very GA-rich spacers.

At this point, we wondered if Nup98 FG domain sequences have anything in common that could possibly explain their unique biophysical properties. To achieve the best possible sampling, we performed exhaustive database searches and identified 666 sequences that matched several stringent criteria for representing Nup98 FG domains (*Figure 10*). Analysis of the sequences indeed revealed several features that appear conserved across all eukaryotic clades, namely: a rather constant number of FG dipeptide motifs per domain (median 43 ± 6), a similar FG motif density (one FG motif per

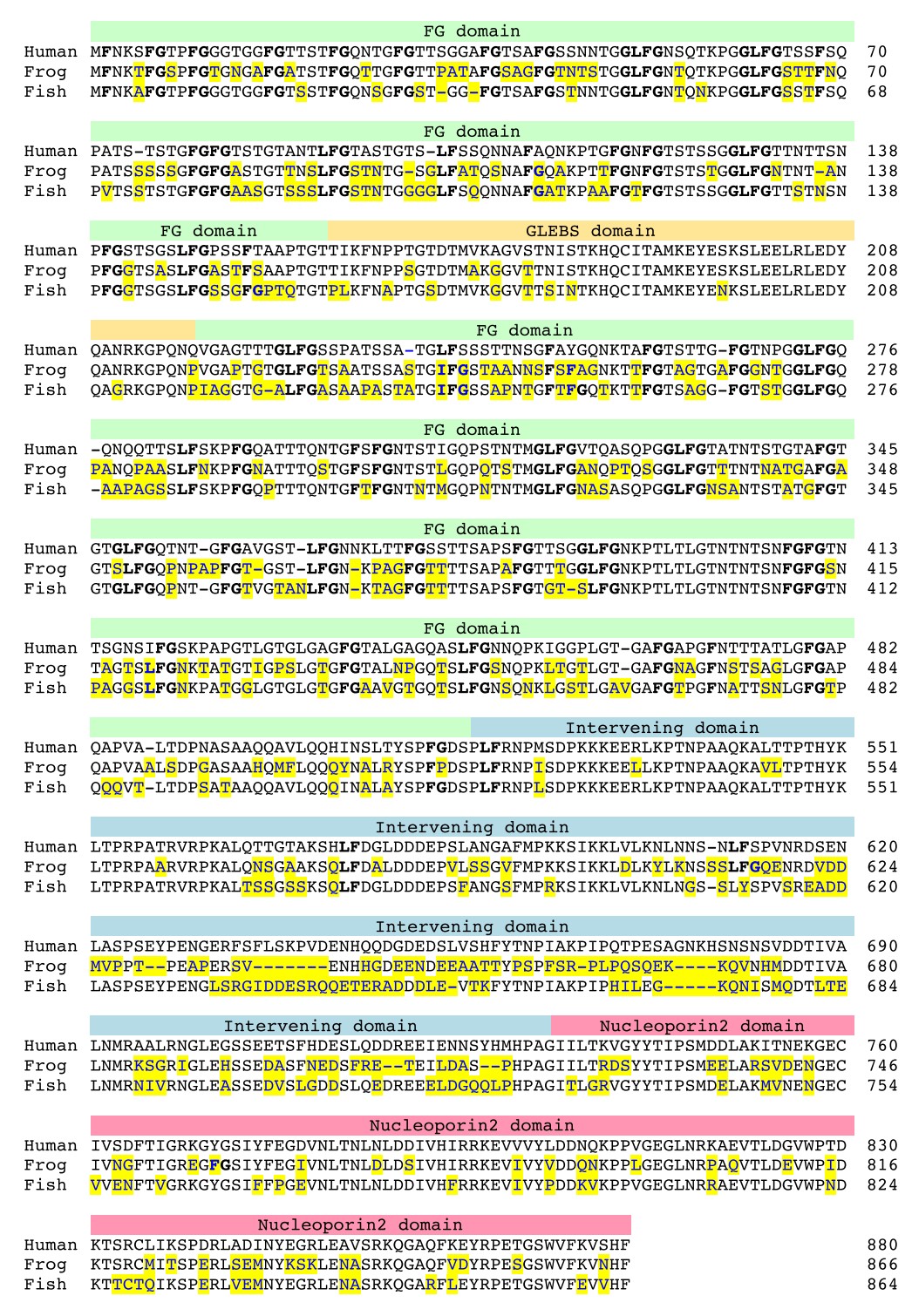

**Figure 9**. High sequence conservation of the Nup98 FG domain amongst vertebrates. Nup98 from human (GI: 530395413), the frog *Xenopus tropicalis* (GI: 523580018) and the fish *Lepisosteus oculatus* (GI: 573879996) were aligned. FG motifs are in bold; deviations from the human sequence are marked in yellow. The domain structure is annotated. Unlike typical intrinsically disordered domains, the Nup98 FG domain shows a similar conservation (≈70% identity) as the globular, folded nucleoporin2 domain (≈75% identity). The majority of exchanges between the FG domains are very conservative, that is mostly permutations between T, S, N, A, and to a lesser extent, with G. Please also note that many exchanges from the human sequence are identical in frog and fish, further supporting the notion of slow evolution. DOI: 10.7554/eLife.04251.012

**Table 2.** Sequence features of the studied FG domains

**FG motif density (occurrence per 100 aa).**

|  | Hs 98 | Bf 98 | Dm 98 | Ce 98 | Sc 100 | Sc 116 | Dd 220 | At 98B | Tt Mac98A | Tb 158 |
|---|---|---|---|---|---|---|---|---|---|---|
| All FG dipeptides | 7.8 | 8.4 | 7.9 | 7.3 | 7.4 | 6.4 | 7.8 | 7.8 | 6.3 | 10.3 |
| (G/A)FG | **2.4** | 1.6 | **2.1** | 0.6 | 0.7 | 1.2 | 0.8 | 1.9 | 0.1 | 1.6 |
| (S/T)FG | 1.2 | 1.9 | 0.7 | 1.0 | 1.2 | 0.2 | 0.3 | 1.2 | 0.4 | 0.2 |
| GLFG | 1.6 | **2.7** | 1.0 | **2.2** | **1.9** | **2.7** | **3.2** | 0.0 | **4.0** | 0.0 |
| SLFG | 0.4 | 0.4 | 0.9 | **2.2** | **2.1** | 0.3 | 1.1 | 0.1 | 0.1 | 0.0 |
| PFG | 0.8 | 0.6 | 0.5 | 0.2 | 0.5 | 0.5 | 1.5 | **1.6** | 0.1 | 0.7 |
| PAFG | 0.0 | 0.0 | 0.7 | 0.0 | 0.0 | 0.1 | 0.0 | **1.2** | 0.0 | 0.0 |
| GFGQ | 0.0 | 0.0 | 0.0 | 0.0 | 0.0 | 0.0 | 0.0 | 0.0 | 0.0 | **7.8** |
| Other FG | 1.4 | 1.2 | 2.0 | 1.1 | 1.0 | 1.4 | 0.9 | 1.8 | 1.6 | 0.0 |

**Bold** data represents dominant FG motifs

**Fraction of hydrophobic and charged residues (w/o GLEBS domains; in %).**

|  | Hs 98 | Bf 98 | Dm 98 | Ce 98 | Sc 100 | Sc 116 | Dd 220 | At 98B | Tt Mac98A | Tb 158 |
|---|---|---|---|---|---|---|---|---|---|---|
| FILVM | 17 | 16 | 16 | 14 | 17 | 15 | 14 | 16 | 16 | 13 |
| FILVMP | 21 | 19 | 22 | 18 | 20 | 18 | 21 | 26 | 20 | 19 |
| DE | 0.4 | 0.0 | 0.5 | 0.0 | 0.2 | 0.0 | 0.0 | 0.3 | 0.0 | 0.2 |
| RK | 1.6 | 1.7 | 1.7 | 1.6 | 2.3 | 1.8 | 0.1 | 0.7 | 0.6 | 1.8 |

**Amino acid composition of the spacer regions (i.e. FG domains w/o FG motifs and GLEBS domains; in %).**

|  | Hs 98 | Bf 98 | Dm 98 | Ce 98 | Sc 100 | Sc 116 | Dd 220 | At 98B | Tt Mac98A | Tb 158 |
|---|---|---|---|---|---|---|---|---|---|---|
| T | **16** | **25** | **14** | 11 | **11** | 9 | **18** | **13** | 12 | 10 |
| S | **11** | 5 | 7 | 13 | **16** | 11 | **11** | **19** | 1 | 3 |
| G | 10 | 13 | 9 | 9 | 9 | 10 | 13 | 6 | **22** | **24** |
| A | 7 | 7 | **16** | 9 | 4 | 7 | 6 | 7 | 10 | **19** |
| N | 6 | 3 | 4 | 10 | **19** | **13** | 2 | 6 | 10 | 1 |
| Q | 5 | 5 | 6 | 8 | **9** | **13** | 10 | 7 | 8 | 11 |
| P | 4 | 3 | 7 | 4 | 3 | 4 | 7 | 9 | 5 | 6 |

**Bold** data represents dominant amino acids

12.5 ± 0.7 residues) and domain-length (median 549 ± 87 residues per FG domain), as well as a very strong bias for G, T, S, A, N, Q, and P in the inter-FG spacers (*Figure 10*).

A comparison of typical intrinsically disordered (IDP) regions (as represented in the DisProt database; *Sickmeier et al., 2007*) and globular proteins (as represented by the PDB database) revealed that Nup98 FG domains are very distinct from these two domain categories (*Figure 11*). They have far fewer charged residues (median 2.5 ± 0.4%) than either average IDP regions (28 ± 7%) or globular proteins (24 ± 3%). They are clearly more hydrophobic than typical IDP regions and, in fact, comparably hydrophobic as globular proteins. Thus, the Nup98 FG domains should have a similar potential to bury hydrophobic side chains from water as globular proteins. In contrast to the latter however, this does not result in a globular fold, but in strong inter-FG repeat cohesion. Taken together, the strong selection against charges and maintenance of hydrophobicity is well in line with our observation that Nup98 FG domains experience water as a "poor solvent" and consequently phase-separate from even rather dilute solutions. The evolutionary conservation of the underlying sequence features suggests again a fundamental functional relevance.

## Effect of hexanediols on FG phase-separation

Early evidence for the importance of hydrophobic contacts in maintaining the permeability barrier of authentic NPCs came from the use of 7% (wt/vol) *trans*-cyclohexane 1,2 diol to interfere with such

**A** Initial BLAST searches with three different Nucleoporin2 domain templates

| Search template (Species/clade/accession number/Nup98 part) | Identified sequences (with E < 1) |
| --- | --- |
| *Homo sapiens* / Opisthokonta / GI:56549643 / aa:678-863 | 1024 |
| *Arabidopsis thaliana* / Plantae / GI:22329468 / aa:887-1027 | 992 |
| *Tetrahymena thermophila* / Chromalveolata / GI:289623519 / aa: 786-923 | 951 |
| Non-redundant sequences: | 1037 |
| Putative Nup98 sequences found in all three searches: | 913 |

**B** Taxonomy report on all identified sequences

| Clade | Opisthokonta | Amoebozoa | Plantae | Chromalveolata | Rhizaria | Excavata |
| --- | --- | --- | --- | --- | --- | --- |
| Number of sequences | 690 | 12 | 145 | 52 | 2 | 12 |

**C** Domain identification and extraction

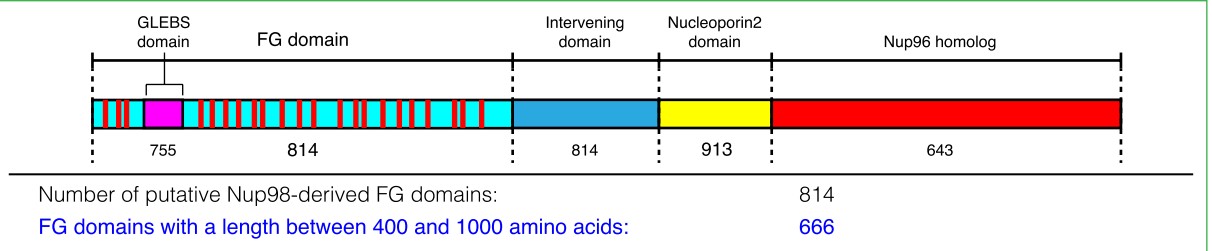

| Number of putative Nup98-derived FG domains: | 814 |
| --- | --- |
| FG domains with a length between 400 and 1000 amino acids: | 666 |

**D** FG dipeptide density

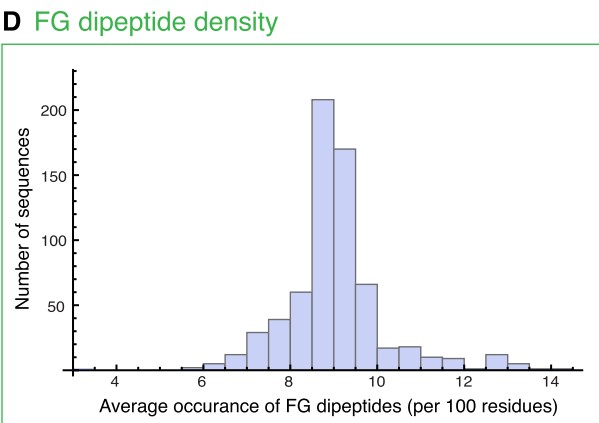

**E** FG domain composition

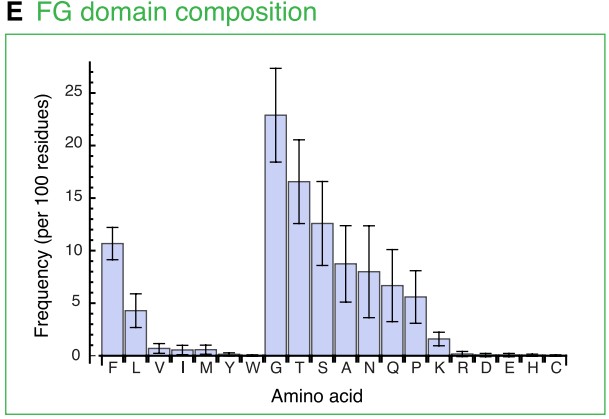

**Figure 10**. Compilation and analysis of a Nup98 ortholog-derived FG domain database. (**A**) The Nucleoporin2 domains of human Nup98/96 (GI: 56549643), *Arabidopsis thaliana* Nup98A (GI: 22329468), and *Tetrahymena thermophila* MicNup98B (GI: 289623519) were used as BLAST templates to identify further Nup98 homologues in the non-redundant NCBI protein database. The 913 sequences identified in all three searches were then analysed further as subsequently described. (**B**) Taxonomic classification of identified full-length Nup98 candidates. (**C**) Cartoon illustrates domain structure of a canonical Nup98-Nup96 fusion protein that includes an FG repeat domain, an embedded Gle2p-binding site (GLEBS domain), an intervening domain, the Nucleoporin2 domain, as well as the Nup96 part. The number of Nup98 candidates that comprise a given module is written underneath. (**D**) FG domains included residues from translation start till the last FG dipeptide but excluded the GLEBS domain (as defined by alignment with the ScNup116 GLEBS domain). For subsequent analyses, only the 666 FG domains with 400–1000 residues were considered. The histogram illustrates the FG dipeptide density distribution with a median of 9 FG dipeptides per 100 residues (or one FG dipeptide per 11 residues). Outliers to higher densities (>10 FG/100 residues) mainly represent domains dominated by less hydrophobic FG motifs (e.g. GFGQ motifs) than the often dominating LFG motifs. (**E**) Average amino acid composition of Nup98 FG domains and corresponding standard deviations. Note that the inter FG spacers are dominated by G, T, S, A, N, Q, and P, while F and L dominate the hydrophobic residues. F shows the smallest coefficient of variation.

interactions in semi-permeabilised HeLa cells. This treatment collapsed the NPC barrier and allowed rapid nucleocytoplasmic equilibration of an MBP reporter, which could be reversed by removal of the reagent (***Ribbeck and Görlich, 2002***). Later, similar effects of n-hexane 1,6 diol on the *S. cerevisiae*

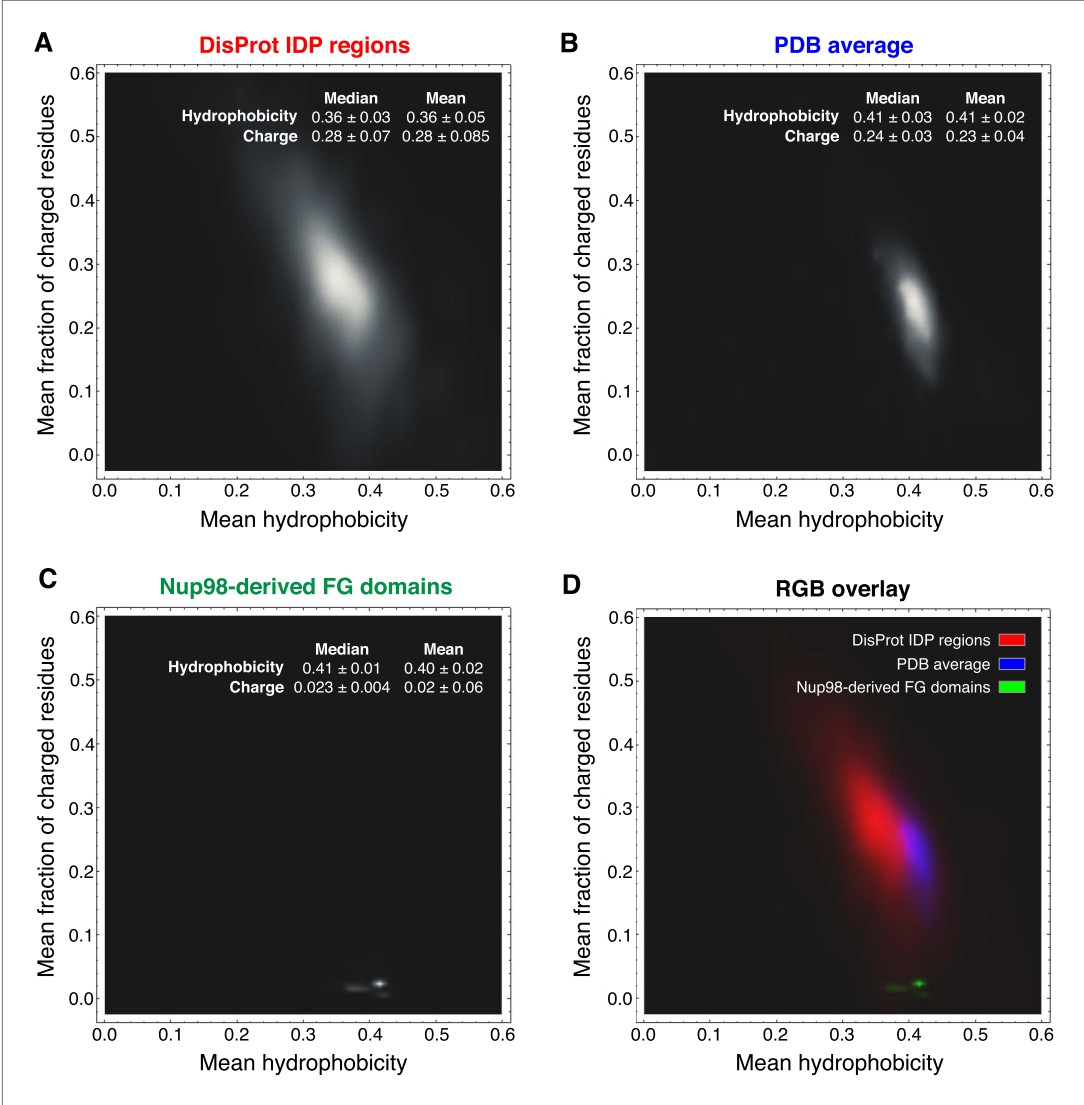

**Figure 11**. Comparison of Nup98 FG domains to intrinsically disordered and globular proteins in terms of charged residue contents and hydrophobicity. Analysis was similar to *Uversky et al. (2000)*, the differences being (i) that we considered not the net charge, but the total fraction of charged residues and (ii) we used a more strictly defined hydrophobicity scale that is not biased by globular protein structures. For each protein sequence, the mean fraction of charged residues was determined by counting D, E, K, and R and dividing this sum by the sequence length. Mean hydrophobicity was calculated according to a scale based on partitioning of Nα-acetyl-amino acid amides between 1-octanol and water at neutral pH (given in *Table 2* in *Fauchere and Pliska, 1983*). For clarity, we re-scaled their numbers linearly to range between 0 and 1, and thus used the following parameters: (R, 0); (K, 0.006); (D, 0.012); (E, 0.113); (N, 0.132); (Q, 0.242); (S, 0.298); (G, 0.31); (H, 0.35); (T, 0.39); (A, 0.405); (P, 0.46); (Y, 0.604); (V, 0.684); (M, 0.687); (C, 0.782); (L, 0.831); (F, 0.859); (I, 0.862); (W, 1). The brightness in the heat maps reflects the number of proteins in a given regime of the plot. (**A**) 667 intrinsically disordered protein (IDP) regions were extracted from the DisProt database (*Sickmeier et al., 2007*) and analysed as described above. Note their wide distribution in the plot, their high content of charges residues and low hydrophobicity. (**B**) Analysis of 34,551 non-redundant protein sequence entries from the PDB (*Bernstein et al., 1978*), representing mostly globular, folded proteins. Note that these sequences are on average less charged and considerably more hydrophobic than the bulk of IDPs. (**C**) Analysis of the 666 identified Nup98 FG domains (excluding the GLEBS domain). Despite also being intrinsically disordered, they strongly cluster in a very narrow region with extremely very low charge density and a hydrophobicity very close to globular proteins. The few outliers with slightly less hydrophobicity represent NQ-rich sequences and reflect the facts (i) that N and Q are more hydrophilic than other typical inter-FG spacer residues (A, T, S, G, P) and (ii) probably that NQ-rich stretches contribute to cohesiveness by conferring very hydrophilic (cross-β) contacts (*Ader et al., 2010*). (**D**) For direct comparison, plots A (in red), B (in blue), and C (in green) were overlaid in a single plot.

NPC barrier were published, whereas ethanol appeared to have little direct consequence on passive nuclear influx of cytoplasmic reporters (*Shulga and Goldfarb, 2003*).

The effect of the hexanediols was attributed to a reversible disruption of inter-FG repeat cohesion (*Ribbeck and Görlich, 2002*; *Patel et al., 2007*). Hence, one would expect that hexanediols should also interfere with the formation of FG phases. *Figure 12* shows that *trans*-1,2-cyclohexanediol or 1,6 hexanediol indeed effectively suppressed the assembly of *S. cerevisiae* Nup116 and *Tetrahymena* Mac98A FG particles, while ethanol had no disruptive effect. Taken together, these findings further affirm the very close relationship between authentic NPCs and the Nup98 FG phases described in this work.

## Large cargo domains require multiple NTRs for efficient partitioning into a selective FG phase

When analysing *Tetrahymena* MacNup98a FG particles in more detail, we noticed that they accumulated NTF2 very efficiently in their interiors, but arrested the Importin β·IBB-GFP complex at their surfaces (*Figure 13A*). As a similar arrest at real NPCs would be very problematic, we decided to dig

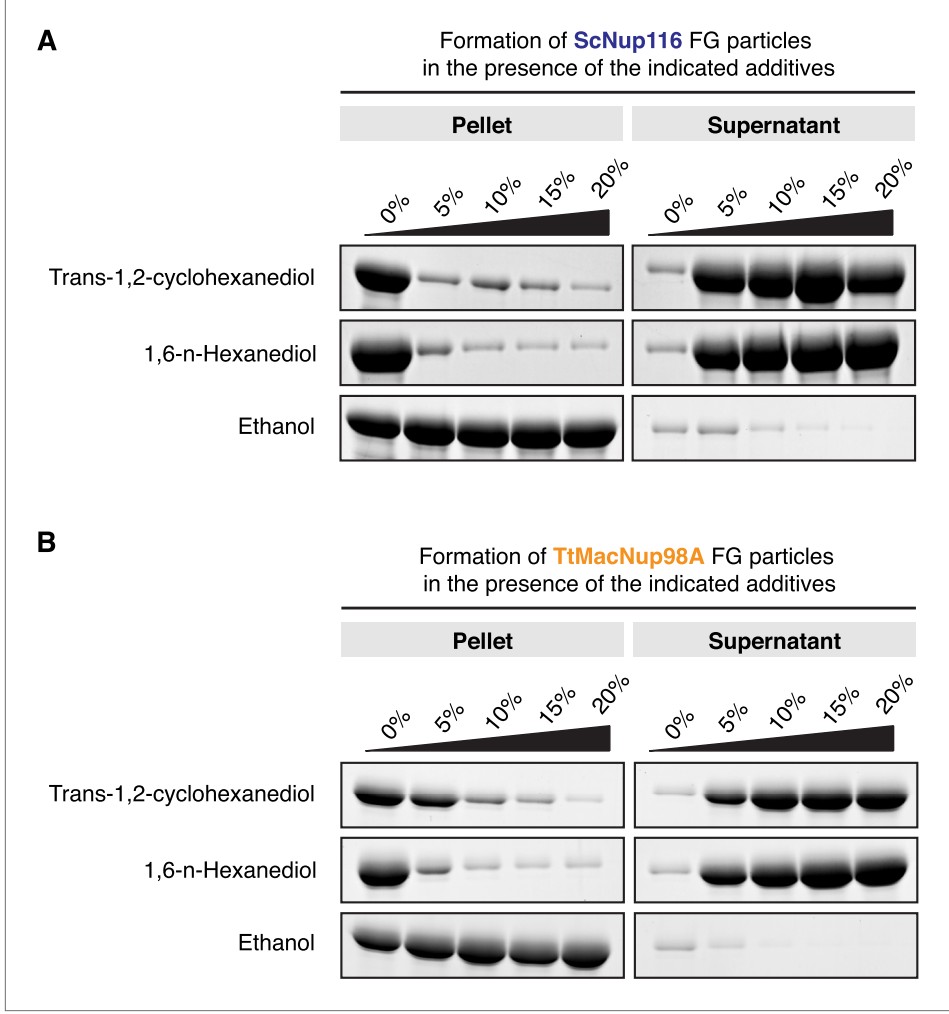

**Figure 12**. Influence of hexanediols on FG particle formation. FG particles were assembled in a volume of 200 µl with 10 µM ScNup116 (**A**) or TtMacNup98A (**B**) FG domain in the presence of increasing amounts of *trans*-1,2-cyclohexanediol, 1,6-hexanediol or ethanol. After 60 min of incubation, formed particles were collected as pellets in a 10-min 20,000×*g* centrifugation step. They were analysed together with the soluble supernatants by SDS-PAGE. Both hexanediols clearly disrupted the FG particles, though TtMacNup98A FG particles appear slightly more resistant than ScNup116 FG particles (consistent with the lower saturation concentration of the Mac98A FG domain). In contrast, ethanol had no disruptive effect, but rather precipitated the FG domains.

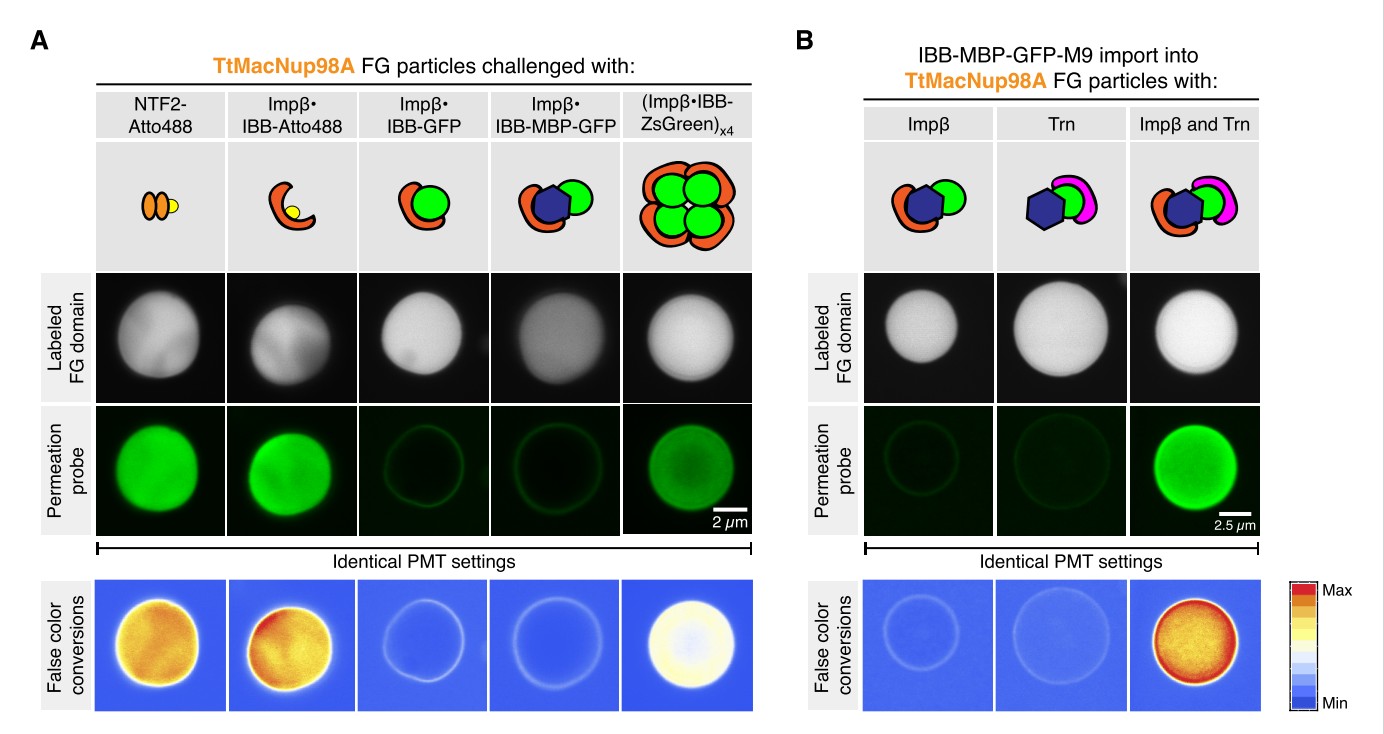

**Figure 13**. Effect of cargo domains and NTR stoichiometry on entry into TtMacNup98A FG particles. FG particles (with 5% Atto390-labeled tracer) were formed with 5 µM TtMacNup98A FG domain. CLSM images show how NTR·cargo complexes of different sizes and NTR-to-cargo ratios partition between FG phase and bulk solvent. IBB (recognized by Importin β) and M9 (recognized by Transportin, Trn) represent two orthogonal nuclear import signals. See **Figure 13—figure supplement 1** and main text for additional information.

The following figure supplement is available for figure 13:

**Figure supplement 1**. Partitioning of various NTR cargo complexes into TtMacNup98 FG particles.

deeper into this problem. At first glance, this was reminiscent of macroscopic *Xenopus* Nup98 FG hydrogels, which were permeable for Importin β-type NTRs only when O-glycosylated with GlcNAc (**Labokha et al., 2013**). However, since there is no indication for a similar modification in ciliates and as the *Tetrahymena* Nup98A domain contains far fewer serines and threonines as potential modification sites than its vertebrate counterpart, we also had to consider other possibilities.

Apart from size, there is actually another difference between NTF2 and the Importin β·IBB-GFP complex: NTF2 is a species with a 'pure' NTR-surface, whereas Importin β also had to 'squeeze' the bound IBB-GFP molecule through the meshes of the gel. It is therefore possible that their spontaneous opening is just too slow or energetically too costly to allow an efficient entry.

If this were true, then one would expect that an NTR·cargo complex with a smaller cargo should not experience this problem. To test this, we replaced the ≈28 kDa GFP moiety by a single Atto488 molecule. This shrunk the cargo to the ≈6 kDa IBB domain, which gets entirely enwrapped by the Importin β molecule leaving no cargo surface exposed (**Görlich et al., 1996**; **Cingolani et al., 1999**). As expected, the Importin β·IBB-Atto488 complex did not remain stuck at the particles' surface but accumulated in their interiors (**Figure 13A**). In parallel controls with the same batch of particles, we observed that the Importin β·IBB-MBP-GFP complex (≈170 kDa) with a further enlarged cargo domain got again firmly arrested at the surface. Much to our surprise, however, we observed that a far larger ≈530 kDa complex, namely a complex comprising four Importin β molecules and an IBB-ZsGreen tetramer, crossed the buffer-particle boundary efficiently and accumulated strongly inside (**Figure 13A** and **13—figure supplement 1**). How can this be explained? ZsGreen/zFP506 is a tetrameric homolog of GFP (**Matz et al., 1999**; **Pletneva et al., 2007**). Consequently, the Importin β·IBB-GFP and Importin β·IBB·ZsGreen complexes have the same NTR:cargo mass ratios, but the tetramerisation

buries much of the otherwise exposed cargo surface that counteracts entry into the FG phase. The tetramerisation should thus allow the Importin to dominate the surface properties of the entire complex, and hence to melt together with the cargo through the meshes of the gel.

At this point, one could still argue that ZsGreen and GFP are different proteins and that differences other than the fraction of exposed cargo surface account for the difference in Importin-mediated particle-entry. In a next step, we therefore used one and the same cargo (an IBB-MBP-GFP-M9 fusion), and just varied the number of bound NTRs. This cargo includes two orthogonal nuclear import signals, the IBB-domain recruiting Importin β (*Görlich et al., 1996*) and the M9-domain conferring nuclear import by Transportin (*Pollard et al., 1996*). *Figure 13B* shows that neither Importin β nor Transportin alone were sufficient to ferry the cargo across the buffer-particle boundary. Yet, when present together on the same cargo molecule, the two importins synergised and mediated efficient influx and strong accumulation of the cargo in the interior of the particles (*Figure 13B*).

Given that the *Tetrahymena* MacNup98A FG particles are particularly tight towards inert macromolecules, it is possible that they exaggerate the inhibitory effect of an exposed cargo domain on barrier-passage. This effect is, however, also evident (with somewhat shifted size limits) for particles from other Nup98 FG domains. *S. cerevisiae* Nup116 FG particles, for example, brightly accumulate NTF2 and the Importin β·IBB-Atto488 complex (*Figure 14A*). The accumulation was already weaker for the Importin β·IBB-GFP complex, while an Importin β·IBB-MBP-GFP complex got stuck at the particles' surface. The larger Importin β·IBB-ZsGreen complex (with its minimized exposed cargo surface)

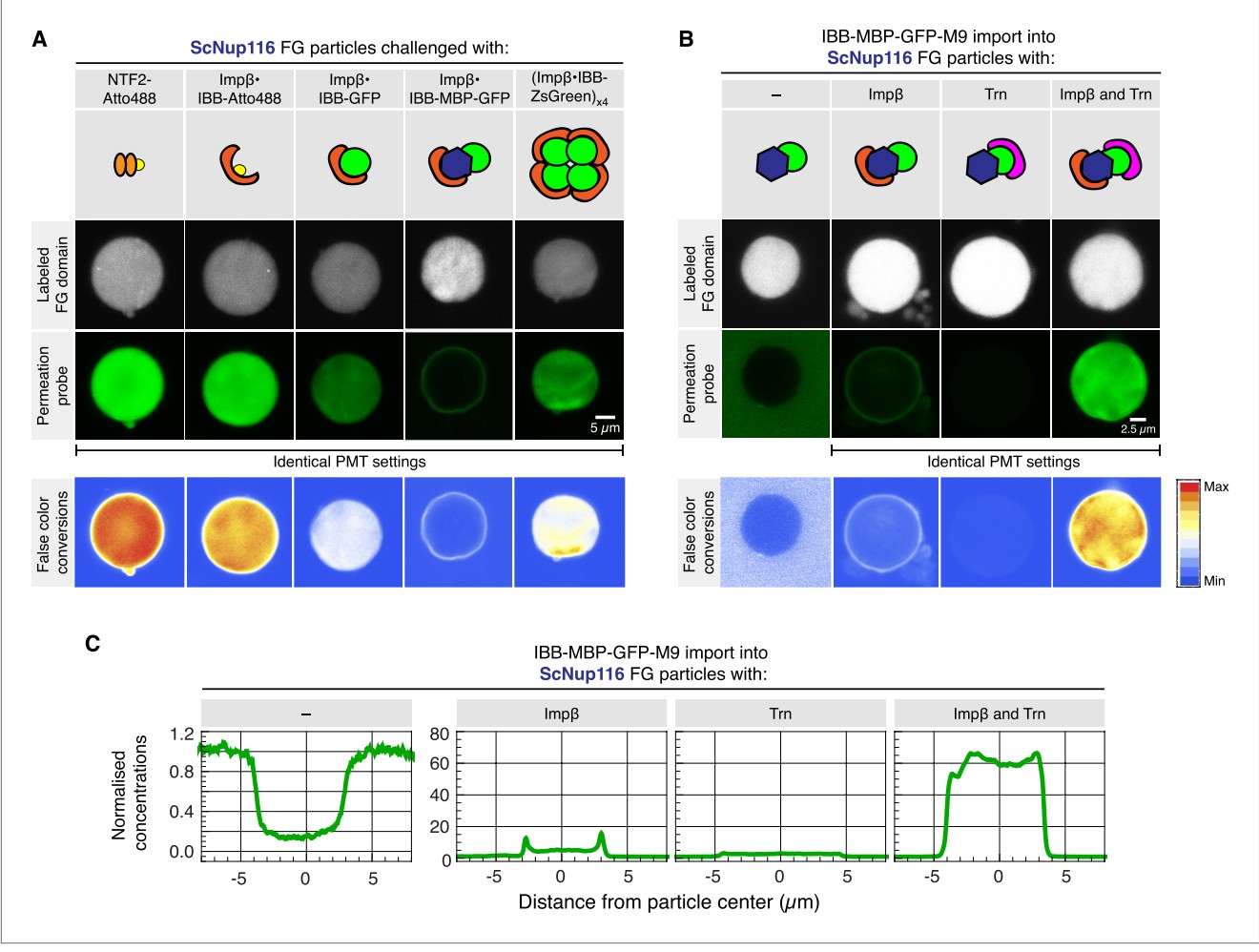

**Figure 14**. Effect of cargo domains and NTR stoichiometry on entry into ScNup116 FG particles. FG particles (with 5% Atto390-labeled tracer) were formed with 10 μM *S. cerevisiae* Nup116 FG domain. See *Figure 13* and main text for additional information.

could again overcome the boundary and accumulated inside. Likewise, the IBB-MBP-GFP-M9 fusion accumulated inside the particles only when Importin β and Transportin were simultaneously bound to this cargo molecule (*Figure 14B*).

Hence, the self-assembled FG phases very nicely recapitulate a key sorting criterion of authentic NPCs, namely that large cargoes require multiple NTRs and thus a good surface coverage by NTRs for an effective pore passage. In fact, the phenomenon of NTR-cooperation was originally observed with a very similar cargo, namely with an IBB-2xMBP-M9 fusion. This cargo bound well to NPCs in the presence of importin β alone, but rapidly traversed NPC only when both, Importin β as well as Transportin, had been recruited (*Ribbeck and Görlich, 2002*). Likewise, *Tu et al., (2013)* recently showed that a single Transportin molecule could confer efficient NPC targeting of a large M9-β-Gal fusion complex, but successful NPC barrier-passage required several Transportin molecules. These remarkable parallels indeed strongly suggest that the permselectivities of in vitro assembled Nup98 FG phases and authentic NPCs are governed by the same mechanistic principles and by functionally equivalent structures.

## Perspectives

We found that Nup98 FG domains assemble from (even dilute) aqueous solutions into very protein-rich phases that reject inert molecules ≥25 kDa in size, but at the same time allow influx of far larger NTR-complexes. They even reproduce the multi-NTR requirement for larger cargoes and recapitulate the kinetics of NPC passage. The most straightforward interpretation of these data is that such phases also assemble inside authentic NPCs and account for the permselectivity of nuclear pores, as previously predicted by the selective phase model.

Yet, the concept of cohesive FG repeat interactions as the basis of NPC transport selectivity has been facing a remarkable resistance from the nuclear transport field over the past 13 years. In light of the available data however, it needs to be emphasised that all suggested alternatives to the cohesion concept appear far more complicated and tied to two rather unlikely assumptions. First, some still elusive process would have to suppress phase-separation and counteract cohesive inter-FG repeat interactions in authentic NPCs. Second, an alternative selectivity mechanism would then have to generate the same permselectivity as observed for self-assembled Nup98 FG phases.

The assembly of selective Nup98 FG phases within NPCs should a priori be a thermodynamically highly favoured process because the local FG motif concentration exceeds the critical concentration at least 100-fold and the system should thus be highly over-saturated with respect to an FG phase-separation. Any block of such an assembly would therefore require a masking of the sticky parts, i.e. of thousands of cohesive FG repeat units per NPC. If effective 'cohesion suppressors' existed, then they would have to be exceedingly abundant NPC ligands, which only leaves NTRs as possible candidates.

We tested this and observed that NTRs do not block Nup98 FG phase-separation even if present in 10-fold molar excess over FG domain (*Figure 15*). This is not surprising because (i) NTR·FG domain interactions are governed by an extreme multivalency (with ≈40 NTR-binding sites on the FG domain and ≈10 FG-binding sites on importin β; *Isgro and Schulten, 2005*), and (ii) such multivalency is prone to drive phase-separation (*Li et al., 2012*)—at least for molar ratios that can be accommodated in the context of NPCs. The tested 10-fold molar excess is probably already far greater than what nuclear pores could possibly hold, because an NPC with 100 anchored FG domains would then be liganded with 1000 NTR molecules (which would at least double the observed mass of an NPC). We consider it, however, possible that NTRs can suppress ectopic FG phase-separation outside NPCs, because there, NTRs are present in an ≈100-fold molar excess over Nup98 FG domains (*Hahn and Schlenstedt, 2011*; *Hülsmann et al., 2012*).

Alternative models for NPC function, avoiding cohesive interactions as a selectivity principle, have been repeatedly proposed. As we discuss in the following, they lack, however, direct experimental evidence or even collide with experimental observations.

The 'virtual gate' model (*Rout et al., 2003*), for example, regards FG domains as entropic brushes that repel inert material while NTRs overcome this entropic barrier by binding to them. This model implies that non-cohesive FG domains are sufficient to create a barrier. This is incompatible with the observation that *Xenopus* NPCs loose their selectivity if the highly cohesive Nup98 FG domain is replaced by non-cohesive ones (*Hülsmann et al., 2012*).

The reversible collapse model extended the virtual gate model by the assumptions that an initially highly extended FG brush collapses or contracts when an NTR binds and that this contraction moves

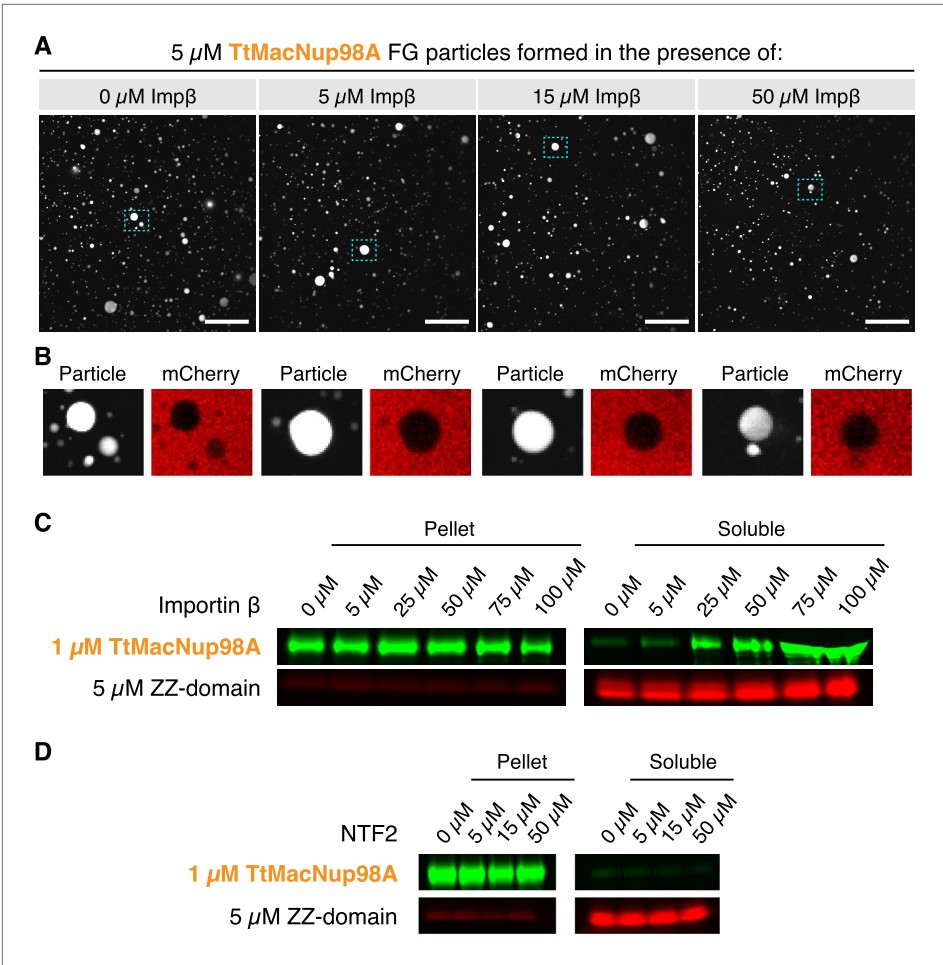

**Figure 15**. Formation of FG particles in the presence of NTRs. (**A**) NTRs do not suppress FG particle formation at molar ratios expected within NPCs. FG particles were formed by dilution of TtMacNup98A to 5 µM (including 5% Atto390-labeled tracer) with Tris-buffered saline containing the indicated concentrations of Importin β. Approximately, 60 s after particle formation, 3 µM mCherry was added. The CLSM images show that FG particles can still form in the presence of a 10-fold molar excess of NTRs. (**B**) An excess of NTRs does not compromise barrier function. Zoom-ins on the particles indicated in **A** show that mCherry is excluded in all cases. (**C**) Importin β increases the critical concentration for FG particle formation only when added in very high excess. FG particles were formed by dilution of TtMacNup98A to 1 µM (including 5% Atto488-labeled tracer) with Tris-buffered saline containing 5 µM Atto565-labeled ZZ-domain and the indicated concentrations of Importin β. After 10 min of incubation, particles were collected by ultra-centrifugation at ≈125,000×*g* for 1 hr, and the amount of FG domains in the pellet and supernatant analysed by SDS-PAGE. For detection of the labelled FG domains, a Fujifilm FLA-9000 fluorescence imager was used. (**D**) Unlike Importin β, NTF2 does not influence the critical concentration of FG particle formation. Experimental setup as in (**C**), with the exception that TtMacNup98A FG particles were formed by dilution with buffer containing the indicated concentrations of NTF2.

the NTR·cargo complex towards the anchor point and eventually across the pore (*Lim et al., 2007*). This model relied on an experiment interpreted such that Importin β collapses the Nup153 FG domain, with a half maximum effect occurring at 30 femtomolar Importin β. Considering that the cellular NTR concentration is 300 million times higher (≥10 µM; *Hahn and Schlenstedt, 2011*), this interpretation, however, implies that those domains are collapsed at all times. More generally, this model is inconsistent with the observation that NPCs still exhibit superb permselectivity when saturated with NTRs (*Ribbeck and Görlich, 2001*; *Yang and Musser, 2006*).

Moreover, neither of the aforementioned alternative models would have predicted that NPC-like permselectivity can be reconstituted with FG domains alone, i.e. independently of any grafting to a channel's inner face. None of them can explain why the propensity of Nup98 FG domains for

spontaneous phase-separation is so perfectly conserved in evolution. And neither can they explain why larger cargoes require multiple NTRs for efficient NPC passage (*Ribbeck and Görlich, 2002*; *Tu et al., 2013*).

The NTR-cooperation effect indeed emphasizes that a simple binding of a translocating species to FG domains is insufficient for facilitated passage, which in fact had been an explicit prediction from the selective phase model (*Ribbeck and Görlich, 2002*). As discussed in the following, the hydrogel model provides two complementary perspectives on this effect. In the first one, we consider the barrier as a 'solvent', into which NTRs, but not inert cargo domains, easily partition. An NTR would improve the cargo's solubility through its own FG affinity as well as by shielding the cargo domain. A complete NTR-assisted partitioning of the cargo into the FG phase would then be an essential stage during a successful NPC passage. Is the cargo, however, too large (in proportion to the NTR), then one would expect 'arrested' intermediates, in which the NTR has already partitioned into the FG phase, while the cargo domain is still excluded, because the energetic penalty is too high for rapidly immersing into the FG phase. Additional NTR molecules should minimise this penalty by further reducing the cargo's accessible surface area and by their own propensity to partition into the FG gel. Indeed, the experiments of *Figure 13* and *Figure 14* visualise such behaviour with remarkable clarity.

The second perspective explicitly considers the inter-FG repeat contacts, which obstruct the passage for inert material, but transiently and locally melt when an NTR binds the corresponding FG motifs. This melting should be restricted to the immediate vicinity of the NTR and thus become inefficient at distant parts of a large cargo. A second NTR molecule acting at this location should then solve this problem.

Notably, the selective phase model per se does not make a prediction as to which cohesive FG domains conduce to barrier formation. Yet, the following observations suggest a prominent contribution of Nup98 FG domains: (i) their extreme propensity to form selective barriers on their own (this study), (ii) their IDP-untypical evolutionary conservation (this study), (iii) their high copy numbers (*Ori et al., 2013*), and (iv) their capacity to restore a selective permeability barrier in FG Nup-depleted NPCs (*Hülsmann et al., 2012*). Other FG domains will, however, also contribute by supplying additional (cohesive or adhesive) FG mass, by forming distinct FG gel layers (of similar or distinct selectivity) and perhaps by forming composite gels with the Nup98 FG domains. Such interplay between distinct FG domains will probably improve the robustness of the system, and it might indeed be an efficient way to fine-tune permselectivity. In the case of *Tetrahymena* we expect, for example, that such blending will relax the extremely tight TtMacNup98A FG hydrogel to a physiological optimum.

Moreover, a perfect control of transport through the NPC requires not only the presence of barrier-forming material, but also that these barrier-elements span the entire cross-section of the central channel and make a tight seal to the scaffold of the pore. This is apparently achieved not only by the direct anchorage of the FG domains, but also by additional interactions between FG repeats and scaffold Nucleoporins, such as yeast Nic96p (*Patel et al., 2007*; *Schrader et al., 2008*), Nup188, Nup192 (*Andersen et al., 2013*), or vertebrate Nup93 (*Xu and Powers, 2013*). The resulting scaffold-FG contacts can thus be seen as a functionally important extension of the cohesive inter-FG meshwork.

## Materials and methods

### DNA constructs, protein expression, and purification

Every protein used in this study was expressed in the *E. coli* strains BLR or NEB Express. Constructs and purification strategies for the following proteins have been described before: Transportin and NTF2 (*Ribbeck and Görlich, 2001*), IBB-MBP-mEGFP, IBB-ZsGreen, mCherry, MBP-mCherry (*Frey and Görlich, 2009*). For all other proteins, new expression vectors have been generated (See *Table 3*). If required, coding sequences were adapted and optimized for expression in *E. coli*. This was particularly important for the FG domain from *Tetrahymena* because this species uses a non-universal genetic code. Sequences and plasmid maps are available on request.

All NTRs, transport substrates and inert molecules were purified by virtue of N-terminal His-tags and native Ni(II) chelate chromatography. Elution was performed with either imidazole or by on-column protease cleavage (*Frey and Görlich, 2014a*, *2014b*). The tags of all imidazole-eluted proteins were cleaved off in solution with TEV protease, proteins were further purified by gel filtration on a Superdex200

**Table 3.** Proteins and corresponding bacterial expression constructs used in this study

| Protein name | Plasmid | Encoding for | Used in figures |
|---|---|---|---|
| HsNup98 FG | pHBS491 | $His_{18}$-HsNup98$_{1-499}$-Cys | 5B, 6, 7 |
| BfNup98 FG | pHBS505 | $His_{18}$-BfNup98$_{1-478}$-Cys | 5B, 6, 7 |
| DmNup98 FG | pHBS503 | $His_{18}$-DmNup98$_{1-580}$-Cys | 5B, 6, 7 |
| CeNup98 FG | pHBS504 | $His_{18}$-CeNup98$_{1-493}$-Cys | 5B, 6, 7 |
| ScNup100 FG | pHBS512 | $His_{18}$-ScNup100$_{2-580}$-Cys | 1B-F, 2, 3A, 5B, 6, 7 |
| ScNup100 FG | pHBS697 | $His_{18}$-ScNup100$_{1-580}$-Cys | 4A, 4C |
| ScNup116$_{\Delta GLEBS}$ FG | pHBS514 | $His_{18}$-ScNup116$_{2-109,167-715}$-Cys | 5B, 6, 7, 12, 13 |
| ScNup116 FG | pHBS698 | $His_{18}$-ScNup116$_{1-736}$-Cys | 4B |
| DdNup220 FG | pHBS241 | $His_{14}$-TEV-DdNup220$_{1-718}$-Cys | 5B, 6, 7 |
| AtNup98B FG | pHBS383 | $His_{14}$-TEV-AtNup98B$_{1-668}$-Cys | 5B, 6, 7 |
| TtMacNup98A FG | pHBS418 | $His_{18}$-TtMacNup98A$_{1-666}$-Cys | 5B, 6, 7, 8, 12, 13, 14, 15 |
| TbNup158 FG | pHBS249 | $His_{14}$-TEV-TbNup158$_{1-565}$-Cys | 5B, 6, 7 |
| ScNsp1$_{274-601}$ FG | pSF654 | $His_{10}$-TEV-ScNsp1$_{274-601}$-Cys | 1B-C |
| ScImpβ | pMR676 | $His_{14}$-brSUMO-ScKap95p | 2, 3A, 4, 13, 14, 15A-C |
| HsImpβ | pICH005 | $His_{14}$-scSUMO-HsImpβ | 5 |
| HsTrn | pKK006 | $His_{10}$-mEGFP-TEV-HsTransportin | 11B, 13B, 14B |
| HsNTF2 | pAL239 | His14-brSUMO-HsNTF2 | 2A, 7, 8, 13A, 14A, 15D |
| ScIBB | pHBS695 | $His_{14}$-ZZ-brNEDD8-Srp1p$_{2-63}$-Cys | 2A, 13A, 14A |
| ScIBB-GFP | pSF807 | $His_{14}$-TEV-ScSrp1p$_{2-63}$-mEGFP | 2A, 3A, 4, 13A, 14A |
| ScIBB-MBP-GFP | pHBS45 | $His_{14}$-TEV-ScSrp1p$_{2-63}$-MBP-mEGFP | 2B, 13A, 14A |
| ScIBB-MBP-GFP-HsM9 | pHBS704 | $His_{14}$-TEV-ScSrp1p$_{2-63}$-MBP-mEGFP-hnRNP A1$_{268-306}$ | 13B, 14B |
| ScIBB-ZsGreen | pSF881 | $His_{14}$-TEV-ScSrp1p$_{2-63}$-ZsGreen | 2A, 13A, 14A |
| ZZ | pHBS237 | $His_{10}$-ZZ-TEV-Cys | 1B, 15C-D |
| mCherry | pSF846 | $His_{14}$-TEV-mCherry | 7, 15B |
| GFP | pHBS349 | $His_{18}$-mEGFP | 5B |
| MBP-mCherry | pSF844 | $His_{14}$-TEV-MBP-mCherry | 2A, 4A-B, 4D, 6 |
| MBP | pSF1911 | $His_{14}$-brSUMO-MBP$_{Gly260Cys}$–$His_6$ | 1E |
| HsActin | pKG017 | $His_{14}$-brSUMO-HsβActin | 4D |

column equilibrated with 44 mM Tris pH 7.5, 290 mM NaCl, 4.4 mM $MgCl_2$, 5 mM DTT, and eventually snap-frozen in liquid nitrogen after addition of 250 mM sucrose.

FG domains were purified using Ni(II) chelate chromatography under denaturing conditions (100 mM Tris pH 8, 8 M GuHCl, 10 mM DTT). Elution was with imidazole in 100 mM Tris pH 8, 20% formamide. If necessary, they were further purified by covalent chromatography, whereby an engineered C-terminal cysteine was allowed to form a disulfide bond with a 2-thiopyridine-activated SH-silica matrix (described below) and elution was achieved by reducing the disulfides with DTT. FG domains were finally re-buffered to 20% acetonitrile, 0.08% TFA and lyophilised. O-GlcNAc modification of FG domains was performed as previously described (**Labokha et al., 2013**).

The 2-thiopyridine-activated SH-silica matrix was produced by the following steps: (i) modifying macroporous silica (Davisil XWP 1000 Å 35–70 micron [Alltech Grom GmbH, Worms, Germany]) with 2% (vol/vol) 3-glycidoxypropyl-trimethoxysilane (CAS#2530-83-8) in xylene o/n @ 60°C, (ii) washing in xylene, ethanol, and finally water, (iii) reaction with aqueous 0.5 M dithiothreitol buffered with 0.1 M Tris/HCl pH 7.5, o/n, 40°C, under argon, (iv) thorough washing in oxygen-free water, transfer in degassed isopropanol:water (50:50) and reaction with 0.1 M 2,2' dithiodipyridine (CAS#2127-03-9), 0.05 M Tris/HCl pH 7.5 in degassed isopropanol:water (50:50) at 20°C under argon. The matrix was stored after washing in 100% degassed isopropanol at 4°C under argon and transferred to binding

buffer immediately prior to use. Covalent chromatography was performed at room temperature. For regeneration, the procedure was repeated at step (iv).

### Estimation of the critical concentrations for FG domain phase-separation

For estimation of the critical concentrations for phase-separation, FG particles were formed by diluting soluble, unfolded FG domain stocks in 2M GuHCl with TBS (50 mM Tris/HCl pH 7.5, 150 mM NaCl) to a range of FG domain concentrations (10 μM, 5 μM, 2.5 μM, 1.25 μM, 0.6 μM, 0.3 μM, 0.1 μM, 0.06 μM). As a first estimate for the saturation concentration, we used the lowest FG domain concentration that yielded detectable particles in the DLS setup (see *Figure 1C*). This number was then refined by measuring the concentration of apparent monomers that co-existed with already formed particles.

Additional method details are given in the legends.

## Acknowledgements

We are grateful to Steffen Frey for sharing plasmids and protein preparations. We thank Jürgen Schünemann for preparative HPLC purification of FG domains, as well as for performing the hexanediol experiment (*Figure 12*) and additional experiments requested by reviewers. We further thank Kevser Gencalp for bacteria with expressed insoluble actin, Indronil Chaudhuri for purified human Importin β, Michael Ridders for the yeast Importin β and Koray Kirli for the transportin expression constructs, Bastian Hülsmann and Steffen Frey for critical reading of the manuscript, as well as the Max-Planck-Gesellschaft for funding.

## Additional information

### Funding

| Funder | Author |
| --- | --- |
| Max-Planck-Gesellschaft (Max Planck Society) | Hermann Broder Schmidt, Dirk Görlich |

The funder had no role in study design, data collection and interpretation, or the decision to submit the work for publication.

### Author contributions

HBS, Conception and design, Acquisition of data, Analysis and interpretation of data, Drafting or revising the article; DG, Conception and design, Analysis and interpretation of data, Drafting or revising the article

## Additonal files

### Supplementary files

• Supplementary file 1. Mathematica program code used for the simulations in *Figure 3*.

• Supplementary file 2. Amino acid sequences of the studied Nup98 FG domains.

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
