## [Decision Letter]

Thank you for sending your work entitled “Nup98 FG domains from diverse species spontaneously phase-separate into particles with exquisite NPC-like permeability” for consideration at *eLife*. Your article has been favorably evaluated by Randy Schekman (Senior editor) and 3 reviewers, one of whom, Karsten Weis, is a member of our Board of Reviewing Editors.

The Reviewing editor and the other reviewers discussed their comments before we reached this decision, and the Reviewing editor has assembled the following comments to help you prepare a revised submission.

This manuscript by Schmidt and Görlich analyzes the characteristics of Nup98 FG-domains purified from multiple species ranging from mammals to ciliates. The authors show that all Nup98 FG-domains tested undergo phase-separation in vitro at relatively low concentrations when diluted out of guanidinium hydrochloride. Furthermore, they also phase-separate spontaneously when expressed in *E. coli*. Importantly, the FG particles that are formed are highly selective and recapitulate key selectivity features of intact NPCs. Using FRAP, they further show that nuclear transport receptors (even with bound cargo) can rapidly diffuse in and out of these particles whereas the FG repeats themselves appear to be immobile.

All reviewers agreed that this manuscript is exciting and contains several novel and important aspects characterizing the behavior and functionality of FG repeats. Therefore there was a consensus that this paper is appropriate for *eLife* after the following revisions.

1) The authors describe the formation of hydrogels from Nup98 monomers as a 'phase transition' but there was concern that this might not be so simple.

How reversible is the formation of the Nup98 hydrogel?

If it is stable, is it indeed due to a phase transition or is there a formation of some stable, for example, amyloid-like structure that provides a framework of stability for the hydrogels?

Do isolated FG bodies shrink with time when the monomer concentration is lowered? If hydrogel formation is irreversible, what stops them from growing further?

If this is due to low monomer concentration, addition of more monomers should lead to larger hydrogels.

Are the FG bodies dissolved by the addition of hexanediol, which was previously shown to inhibit NPC transport and interfere with homotypic FG interactions?

The authors should add some, more quantitative phase separation diagrams to measure at what concentration phase separation occurs.

Furthermore, they should calculate the mass of material as monomers and as hydrogels for a range of initial concentrations. If there is a critical concentration, the total monomer concentration should be constant.

The authors argue that transport receptors do not interfere with FG body assembly even at high concentration because of the multivalency of their interactions with FG repeats.

What about NTF2? The presence of NTRs should influence the critical concentration that is needed to phase separate and, for example, low-valence NTRs should interfere whereas multivalent NTRs should allow for separation at lower concentrations (see e.g. [39]).

If NTRs do not affect hydrogel formation wouldn't that argue that critical steps in the formation of the hydrogel do not rely on FG–FG interactions?

2) The authors should tone down the conclusion on the exclusive role of Nup98's FG repeat domain in selectivity. Clearly, there are other FG domains within the NPC that are functionally important and the 'perspective' section should include a broader discussion of the various types of FG domains. Do the authors think pure Nup98 hydrogels will form in the pore as well, by excluding FG domains of different origin? One would expect that at least the cohesive FGs would all interact with each other, if sterically allowed. Do the authors expect distinct FG-zones of distinct character within the NPC?

---

## [Author Response]

*1) The authors describe the formation of hydrogels from Nup98 monomers as a 'phase transition' but there was concern that this might not be so simple*.

We originally adopted the term “phase transition” from the RNA granule field, where it is widely used to describe a process that is, strictly speaking, a phase separation in a biological system or, more specifically, a liquid–liquid unmixing leading to the formation of RNA granules or RNA-protein or just protein-rich droplets. We agree that the term is problematic, in particular as it originates from physics with a different meaning: “A phase transition is the transformation of a thermodynamic system from one phase or state of matter to another one by heat transfer”. In analogy to the biophysics of lipid bilayers, the process we studied would also be a phase separation and not a phase transition.

Given this potential for confusion, we replaced the term “phase transition” in our manuscript by “phase separation”, because this is what we meant and what is immediately obvious from the microscopic images showing protein-rich FG phases clearly separated from the surrounding buffer.

How reversible is the formation of the Nup98 hydrogel?

Again, here is a potential for confusion, because the term “reversibility” is used differently in classic thermodynamics and in biology. Thermodynamics considers changes to closed systems that can be reversible or irreversible. If we started with an oversaturated solution of FG domain protomers, and phase separation and the formation of FG particles occurred, then this would be thermodynamically irreversible, because there would be no spontaneous return to the initial state. However, biochemistry would nevertheless consider the endpoint to be a reversible steady state if protomers from the particles are in steady exchange with protomers in the surrounding buffer. Here, the classification as “reversible” or “irreversible” is also a pragmatic issue of considered time scales.

As far as we can tell, the formation of FG particles is reversible according to the biochemical perception. We include two additional experiments to support our view. In the first one, we formed FG particles with 1, 3, and 10 µM Nup116 FG domains. We observed that the amount of particulate FG domains increased with protomer concentration, while the concentration of remaining soluble FG domain stayed constant (Figure 16). This is the expected result if the distribution of the FG domain between protein-rich phase and buffer is governed by a reversible equilibrium state.Author response image 1.Formation of ScNup116 FG particles at different concentrations.

In a second experiment, we first formed FG particles at 10 µM protomer concentration. One sample was subsequently left at 10 µM, while the others were diluted to 1 or 3 µM final protein concentration. After an extended incubation period (4 hr), we observed that the concentration of the soluble Nup116 FG domain was identical in all three samples, while the ratios of particulate: soluble matter dropped with decreasing protein concentration (Figure 17). This is expected when the FG domain can indeed reversibly re-equilibrate between particles and bulk buffer.Author response image 2.Reversibility of FG particle formation.

It should be said though that a phase separation is not necessarily defined through reversibility. There are many examples where the separation is irreversible (for instance the unmixing of a melt during solidification).

If it is stable, is it indeed due to a phase transition or is there a formation of some stable, for example, amyloid-like structure that provides a framework of stability for the hydrogels?

As mentioned above, we abandon the term “phase transition” because it is very prone to confusion when used to describe something that is actually an unmixing event.

Extending the question to “phase separation”, we see no contradiction between a structure being formed by a phase separation and the same structure being stable (kinetically and/or thermodynamically). Likewise, a structure can be stable over time and yet exchange constituents with the surrounding buffer.

Selective FG hydrogels may or may not contain amyloid-like, i.e. cross-beta sheet, structures. This is a major point in the present and two previous studies. The NQ-rich FG domains from *S. cerevisiae* Nup100 or Nup116 form FG particles that stain positive with ThioflavinT and, by this criterion, contain such structures. Using solid-state NMR, we previously detected cross-β structures in FG hydrogels derived from the similarly NQ-rich N-terminus of Nsp1p (1). Yet, other FG hydrogels appear to lack such structures, as e.g. judged from the very weak ThioflavinT-staining of the *Tetrahymena* MacNup98A FG particles. Again, previous solid-state NMR data second this notion, as FG hydrogels derived from the fully GlcNAc-modified *Xenopus* Nup98 FG domains clearly lack cross-β structures (38).

Do isolated FG bodies shrink with time when the monomer concentration is lowered?

Yes. They eventually disappear when we keep removing monomers from the equilibrium, for example by repeated steps of pelleting and resuspending in fresh buffer. We mention this as technical issue in the text, because an excessively thorough purification of FG inclusion bodies by too many centrifugation steps may lead to a complete loss of material, in particular for FG domains with a higher critical/ saturation concentration. This applies e.g. to the Nup116 FG domain (c ≈ 0.7 µM), but is less of an issue for the TtMac98A FG domain (c ≈ 0.02 µM). Please also refer to the aforementioned Figure 17.

*If hydrogel formation is irreversible, what stops them from growing further? If this is due to low monomer concentration, addition of more monomers should lead to larger hydrogels*.

Monomers become limiting (also in case of an irreversible assembly). Of course, one can play games like enlarging pre-existing FG particles, couple such enlargement with a switch in colors etc. But considering the length of the manuscript and the number of figures, we prefer to leave this for a future study.

Are the FG bodies dissolved by the addition of hexanediol, which was previously shown to inhibit NPC transport and interfere with homotypic FG interactions?

Indeed, our lab originally discovered that hexanediols renders HeLa NPCs non-selectively permeable (56). The Goldfarb lab later confirmed this for yeast NPCs (64). We have now tested the effect of trans-1, 2-cyclo-hexanediol and 1,6-n-hexanediol on FG particle formation and observed that these agents also interfere with FG phase formation in our in vitro system (see new Figure 12). This finding is consistent with the original idea that hexanediols disrupt the FG–FG interactions within NPCs and thus increase their permeability. It moreover highlights the unique resemblance of the NPC barrier to the self-assembled FG phases that we report on.

*The authors should add some, more quantitative phase separation diagrams to measure at what concentration phase separation occurs*.

It is unclear what the reviewers meant with “phase separation diagrams”. In classic thermodynamics, these are (typically two-dimensional) representations of phase transitions as functions of pressure and temperature; in the fields of detergents, this is often used to represent transitions between monomers, micelles, and inverted micelles as a function of temperature and ionic strength.

For all 10 FG domains discussed in this study, we tested entire series of domain concentrations to estimate the lowest one that would allow phase separation. These numbers are listed as “estimated critical concentrations” in Table 1. They are certainly accurate enough to support the argument that the expected local Nup98 FG domain concentration in assembled NPCs exceeds the critical concentrations for phase separation by a very large factor.

For these measurements, we used standardised conditions (21°C, physiological pH and ionic strength) and cannot see why filling phase diagrams with multidimensional titration data should be required for any argument we make.

*Furthermore, they should calculate the mass of material as monomers and as hydrogels for a range of initial concentrations. If there is a critical concentration, the total monomer concentration should be constant*.

This is true, and indeed we did measure entire concentration series to derive the listed critical (saturation) concentrations. This is now written more clearly in the Methods.

In addition, we now included Figures 16 and 17 to document for the Nup116 domain that FG particles are indeed in equilibrium with a constant monomer concentration (see above).

*The authors argue that transport receptors do not interfere with FG body assembly even at high concentration because of the multivalency of their interactions with FG repeats*.

We write that a 10-fold molar excess of importin β does not block the FG body assembly. We now also include a direct solubility test to document this (new Figure panels 15C and 15D). At a ≥10-fold molar excess, a slight increase in solubility becomes evident. However, this would already mean 1000 NTR molecules being simultaneously bound to a single NPC, while ≈100 would be the expected order of magnitude. We extended the Discussion to make this even clearer.

*What about NTF2? The presence of NTRs should influence the critical concentration that is needed to phase separate and, for example, low-valence NTRs should interfere whereas multivalent NTRs should allow for separation at lower concentrations (see e.g.*
[39]*)*.

NTF2 does not make the Mac98A FG domain more soluble, even when used at a 50-fold molar excess (see new Figure 15).

If NTRs do not affect hydrogel formation wouldn't that argue that critical steps in the formation of the hydrogel do not rely on FG–FG interactions?

We did not write that NTRs do not affect phase separation. They do not block the process at the stoichiometries that are allowed by the volume constraints of an NPC.

Moreover, previous data strongly argue for a critical contribution of FG–FG interactions: mutating FG to e.g. SG or AG motifs completely abolishes cohesive interactions (23; 47; 19; 1; 30).

2) The authors should tone down the conclusion on the exclusive role of Nup98's FG repeat domain in selectivity. Clearly, there are other FG domains within the NPC that are functionally important and the 'perspective' section should include a broader discussion of the various types of FG domains. Do the authors think pure Nup98 hydrogels will form in the pore as well, by excluding FG domains of different origin? One would expect that at least the cohesive FGs would all interact with each other, if sterically allowed. Do the authors expect distinct FG-zones of distinct character within the NPC?

All experimental data in this study were obtained for Nup98 FG domains, so it is natural that these domains are the focus of any discussion. In addition, we extensively addressed and discussed the issue of distinct FG zones in an earlier publication (38). We feel that the arguments given there are still fully valid, but also that it is not an option to replicate an earlier Discussion section in a new publication. Instead, we now write:

“Notably, the selective phase model per se does not make a prediction as to which cohesive FG domains conduce to barrier formation. Yet, the following observations suggest a prominent contribution of Nup98 FG domains: (*i*) their extreme propensity to form selective barriers on their own (this study), (*ii*) their IDP-untypical evolutionary conservation (this study), (*iii*) their high copy numbers (45) and (*iv*) their capacity to restore a selective permeability barrier in FG Nup-depleted NPCs (Hülsmann et al., 2013). Other FG domains will, however, also contribute by supplying additional (cohesive or adhesive) FG mass, by forming distinct FG gel layers (of similar or distinct selectivity) and perhaps by forming composite gels with the Nup98 FG domains. Such interplay between distinct FG domains will probably improve the robustness of the system and it might indeed be an efficient way to fine-tune permselectivity. In the case of *Tetrahymena* we expect, for example, that such blending will relax the extremely tight TtMacNup98A FG hydrogel to a physiological optimum.”